# Bounded Model Checking for Metric Temporal Logic Properties of Timed Automata with Digital Clocks [note 1]

**DOI:** 10.3390/s22239552

**Published:** 2022-12-06

**Authors:** Agnieszka M. Zbrzezny, Andrzej Zbrzezny

**Affiliations:** 1Faculty of Mathematics and Computer Science, University of Warmia and Mazury, Sloneczna 54, 10-710 Olsztyn, Poland; 2Department of Mathematics and Computer Science, Jan Dlugosz University in Czestochowa, Armii Krajowej 13/15, 42-200 Czestochowa, Poland

**Keywords:** bounded model checking, timed automata, digital clocks, metric temporal logic, satisfiability problem

## Abstract

Metric temporal logic (MTL) is a popular real-time extension of linear temporal logic (LTL). This paper presents a new simple SAT-based bounded model-checking (SAT-BMC) method for MTL interpreted over discrete infinite timed models generated by discrete timed automata with digital clocks. We show a new translation of the existential part of MTL to the existential part of linear temporal logic with a new set of atomic propositions and present the details of the new translation. We compare the new method’s advantages to the old method based on a translation of the hard reset LTL (HLTL). Our method does not need new clocks or new transitions. It uses only one path and requires a smaller number of propositional variables and clauses than the HLTL-based method. We also implemented the new method, and as a case study, we applied the technique to analyze several systems. We support the theoretical description with the experimental results demonstrating the method’s efficiency.

## 1. Introduction

This paper is a full version of the extended abstract published in informal proceedings of the 25th International Workshop on Concurrency, Specification and Programming [1]. Improvements and extensions compared to that paper are listed in Appendix A.

There are many ways to check the model. Hardware- and software-based systems are increasingly used in safety-critical situations to control and connect physical systems, e.g., simple cardiac pacemakers or very complex space rockets. The complexity and criticality of systems also increase the need for effective verification techniques.

The specification language and the system model usually depend on the type of property we want to verify. Suppose we want to verify the discrete-time execution properties of a system. In that case, discrete-time logic may be the correct specification choice, and the system may best be represented as a discrete timed automaton.

Integrating specialized components is fundamental to many embedded engineering projects. The behavior of these components is often specified in informal timing diagrams that engineers interpret during interface hardware and software design [2,3,4,5,6,7]. The timed automata are one of the models that enable the formal modeling of those components.

Timed automata (TA) [8] are finite-state automata augmented with a finite set of variables called clocks. The clocks are used to measure the elapsed time. Timed automata are very convenient for modeling and reasoning about timed systems: they combine a powerful formalism with advanced expressiveness and efficient algorithmic and tool support. The timed automata formalism is applied to the analysis of software and asynchronous circuits [9] and real-time control programs [10].

The model-checking technique is widely used in sensor verification [2,3,4,5,6,7]. The sensor networks are modeled, e.g., by the network of timed automata, and their properties are specified in terms of temporal logic.

One of the most famous frameworks in the specification and verification of computer systems is temporal logic. There are many types of temporal logic to express the requirements of the systems: computation tree logic (CTL) [11], soft real-time CTL (RTCTL) [12], linear temporal logic (LTL) [13], and metric temporal logic (MTL) [14].

Linear temporal logic (LTL) allows expressing properties about each execution of a system, such as the fact that any occurrence of a problem eventually triggers the alarm. Metric temporal logic (MTL) extends LTL by constraining the temporal operators with time intervals. It was introduced by Koymans [14] in 1990 and has appeared as a real-time specification formalism. MTL has two main semantics: “continuous” and “pointwise” [15,16,17,18]. The pointwise semantics is based on timed words, the widespread interpretation for systems modeled as timed automata [8]. Both semantics have been extensively studied [15,16,17,18]. MTL allows expressing, for example, that any occurrence of a problem in a system will trigger the alarm within at most five units of time. Here, we consider MTL with pointwise semantics interpreted over linear discrete infinite digital-clock models [19] generated by timed automata with integer time.

Bounded model checking [20,21,22] (BMC) is one of the symbolic model-checking techniques designed for finding witnesses for existential properties or counterexamples for universal properties. Its main idea is to consider a model reduced to a specific depth. The method works by mapping a BMC problem to the satisfiability problem (SAT). The MTL satisfiability and model-checking problems are undecidable over interval-based semantics [23]. It has led to various restrictions being considered on MTL to recover decidability [24,25].

We provide a new, efficient method of the bounded model-checking technique for metric temporal logic properties of timed automata with digital clocks, which can be successfully used to verify sensor networks.

Developing new model-checking techniques for a network of automata is an essential research direction. It is due to the fact that timed automata are used to verify the modes of, often, life-critical time systems. Systems and their properties are becoming more and more complex. It results in developing model-checking methods that will work faster than the older methods. To be successfully applied, these methods should be faster than the old ones and use as little memory as possible. However, they should also be easy to implement and understand so that everything is evident at the design stage of such a method. For such a reason, in this paper, we developed a completely new and much faster method of bounded model checking than the one presented in [26].

The main contributions of the paper are as follows:Defining the translation of the existential model-checking problem for MTL to the existential model-checking problem for linear temporal logic with additional propositional variables $q_{I}$ (this logic is denoted by ${\mathrm{LTL}}_{q}$);Clarification of the steps of the new method;Proving the correctness of the above translation;Defining bounded semantics for ${\mathrm{LTL}}_{q}$;Defining the BMC algorithm;Implementing the new method;A detailed experimental evaluation of the old and the new methods on two earlier presented benchmarks: a timed generic pipeline paradigm (TGPP) and a timed train controller system (TTCS),Modeling a dining philosophers problem with time as the timed dining philosophers problem (TDPP);A detailed experimental evaluation of the old and the new methods on TDPP.

In this paper, we used the weakly monotonic semantics [27] for timed automata with digital clocks [19,27]. The main steps of our new method for MTL and TA with discrete time can be described as follows: first, the infinite timed model is reduced to a finite model. Next, the MTL formulae are translated to ${\mathrm{LTL}}_{q}$ formulae [1], and eventually, since the interval modalities in MTL appear as literals in the ${\mathrm{LTL}}_{q}$ formula, existential properties are reduced to a satisfiability problem (SAT). Our method’s main advantages are that the translation from MTL to ${\mathrm{LTL}}_{q}$ requires neither new clocks nor new transitions. Moreover, our BMC method needs only one path, whereas the BMC method from [26] needs many paths depending on a given formula φ. Thus, one may expect that our method is much more effective since the intuition is that an encoding that results in fewer variables and clauses is usually easier to solve.

We evaluate the BMC method using a timed generic pipeline paradigm (TGPP), a timed train controller system (TTCS), and the timed dining philosophers problem (TDPP), which we model by a network of discrete timed automata and compare with the corresponding method [26].

### Related Work

Timed automata were introduced in the early 1990s by Alur and Dill [8] to model real-time systems. Timed automata cause the specification and verification of models of real-time systems to be easier. The two primary semantics are discussed in the literature: the discrete-time and dense-time semantics [8]. However, the dense-time semantics is more natural from a real-life point of view. It allows us to model real-time systems easily.

Our choice of time domain is N, the set of natural numbers. In our method, the key property of the time domain is its discreteness, which implies that a finite amount of events can happen at different times in any interval of nonzero length. There are many methods for verifying real-time systems using discrete-time models [12,28,29,30,31]. Authors of [19] established that the timed reachability problem has the same answer, irrespective of the choice between N and IR under certain restrictions.

The other formalisms for discrete time modeling apart from discrete timed automata were presented, such as durational transition graphs (DTG [32]) and embedded system modeling language (EMLAN [33]).

Discrete time models were also widely used for modeling systems’ behaviors [34,35,36,37,38].

MTL has been widely discussed in the literature. Checking properties expressed in MTL in timed automata is still an actual research topic [39,40,41,42,43,44]. In [45], the authors took into account MTL over the N. They also used the pointwise semantics over the N and considered two semantic variants: the non-strict and strict semantics. They devised two translations from MTL to LTL:The time difference translation for strict semantics, where new propositional variables encode time differences between states (the time difference translation is similar to the method presented in [1]).The gap translation for the strict semantics uses a new propositional variable, called gap, to encode the jumps between states. The gap is intended to be true in LTL states corresponding to unmapped time points in MTL models. The main idea for their translation is to map each timed state sequence into a state sequence. Both LTL translations are exponential in the size of the MTL input formula due to the binary encoding of the numbers in the intervals.

In [26], the authors investigated a SAT-based BMC method for MTL that is also interpreted over linear discrete infinite time models generated by discrete timed automata. They translated the existential model-checking problem for MTL into the existential model-checking problem for a variant of linear temporal logic (called HLTL). They also provided a SAT-based BMC technique for HLTL. The presented translation requires as many new clocks as there are intervals in a given formula. It also requires adding exponential resetting transitions and many paths that depend on a given MTL formula. The complexity of the satisfiability and model-checking problems for fragments of MTL concerning different semantic models were studied in [46] and in [15]. MTL expressiveness was extensively discussed in [17,47,48]. The BMC problem for MTL properties of timed automata with the dense time was discussed in [49]. However, experimental results have shown that this is not feasible.

Additionally, other types of logic were used for the specification of discrete-time systems, such as QsCTL [29], which extends CTL [11] with quantitative bounded temporal operators and is a variant of RTCTL [11], discrete-time CTL [33], and HyperLTL [50], which is a temporal logic for hyper-properties, which allows reasoning about multiple execution paths simultaneously.

## 2. Discrete Timed Automata and MTL

In this paper, we used the weakly monotonic semantics [27] for timed automata with digital clocks [19,27]. The paper [19] shows that bounded invariance MTL properties and bounded-response MTL properties are digitizable. That is why we consider timed automata with digital clocks.

The formalism of timed automata was defined in [8] by Alur and Dill for representing systems with real-time constraints. A timed automaton is a finite automaton which manipulates finitely many variables called clocks.

### 2.1. Discrete Timed Automata

Let N={0,1,2,…} be the set of natural numbers, and N+=N\{0}. We assume a finite set X={x0,…,xn−1} of variables, called *clocks*. Each clock is a variable ranging over N. A *clock valuation* is a total function v:X→N that assigns to each clock *x* a non-negative integer value v(x). The set of all the clock valuations is denoted by NX. For X⊆X, the valuation v′=v[X:=0] is defined as ∀x∈X, v′(x)=0 and ∀x∈X\X, v′(x)=v(x). By δ, we denote a delay of δ>0 time units. For δ∈N+, v+δ denotes the valuation v″ such that ∀x∈X,v″(x)=v(x)+δ; let x∈X, c∈N, and ∼∈{<,⩽,=,⩾,>}. The set C(X) of *clock constraints* over the set of clocks X is defined by the abstract grammar:

$$\begin{array}{c} \mathrm{cc}:=x∼c\;|\;\mathrm{cc}\wedge \mathrm{cc}. \end{array}$$
Let *v* be a clock valuation, and cc∈C(X). A clock valuation *v* satisfies a clock constraint cc, written as v⊨cc if cc evaluates to true using the clock values given by the valuation *v*.

 **Definition 1.** 
*A discrete timed automaton (DTA for short) is a tuple*


$$A=(Act,Loc,\ell^{0},X,T,Inv,\mathrm{AP},V),\mathrm{where}$$


*Act is a finite set of actions,*

*Loc is a finite set of locations,*

*$\ell^{0}\in Loc$ is the initial location,*

*X is a finite set of clocks,*

*$T\subseteq Loc\times Act\times C\left(X\right)\times 2^{X}\times Loc$ is a transition relation,*

*Inv:Loc→C(X) is a state invariant function,*

*AP is a set of atomic propositions, and *

*$V:Loc\to 2^{\mathrm{AP}}$ is a valuation function assigning to each location a set of atomic propositions true in this location.*


*Each t∈T, denoted by $\ell \overset{\sigma ,\mathrm{cc},X}{⟶}\ell^{\prime }$, represents a transition from ℓ to $\ell^{\prime }$ on the action σ. X⊆X is the set of the clocks to be reset upon this transition, and cc∈C(X) is the enabling condition for t.*


### 2.2. Product of a Network of Discrete Timed Automata

A network of discrete timed automata can be composed into a global (*product*) discrete timed automaton [51] in the following way. The transitions of the discrete timed automata that do not correspond to a shared action are interleaved, whereas the transitions labeled with a shared action, are synchronized.

Let n∈N, I={1,…,n} be a non-empty and finite set of indices, $\left\{A_{i}|i\in I\right\}$ be a family of discrete timed automata $A_{i}=\left(Act_{i},Loc_{i},\ell_{i}^{0},X_{i},T_{i},{Inv}_{i},{\mathrm{AP}}_{i},V_{i}\right)$ such that $X_{i}\cap X_{j}=\emptyset$ and ${\mathrm{AP}}_{i}\cap {\mathrm{AP}}_{j}=\emptyset$ for i≠j. Moreover, let $I\left(\sigma \right)=\left\{i\in I|\sigma \in Act_{i}\right\}$. The *parallel compositioni* of the family $\left\{A_{i}|i\in I\right\}$ of discrete timed automata is the discrete timed automaton A = $\left(Act,Loc,\ell^{0},X,T,Inv,\mathrm{AP},V\right)$ such that:$Act=\prod_{i\in I}\left(Act_{i}\right)$,$Loc=\prod_{i\in I}Loc_{i}$,$\ell^{0}=\left(\ell_{1}^{0},\ldots ,\ell_{n}^{0}\right)$,$X={⋃}_{i\in I}X_{i}$,$Inv\left(l_{1},\ldots ,l_{n}\right)={⋀}_{i=1}^{n}{Inv}_{i}\left(l_{i}\right)$,$\mathrm{AP}={⋃}_{i\in I}{\mathrm{AP}}_{i}$,$V\left(\ell_{1},\ldots ,\ell_{n}\right)={⋃}_{i=1}^{n}V_{i}\left(\ell_{i}\right)$
and a transition is defined as follows:
$$\begin{array}{c} (\left(\ell_{1},\ldots ,\ell_{n}\right),\left(\sigma_{1},\ldots ,\sigma_{n}\right),{⋀}_{i\in I}{\mathrm{cc}}_{i},{⋃}_{i\in I}X_{i},\left(\ell_{1}^{\prime },\ldots ,\ell_{m}^{\prime }\right))\in T\\ \mathrm{iff}(\forall i\in I))(\ell_{i},\sigma_{i},{\mathrm{cc}}_{i},X_{i},\ell_{i}^{\prime })\in T_{i})\mathrm{and}(\forall i\in I\backslash I\left(\sigma \right))\ell_{i}^{\prime }=\ell_{i}. \end{array}$$

### 2.3. Concrete Model

The semantics of the DTA is defined by associating with it a transition system, which we call a *concrete model*.

 **Definition 2.** 
*Let $A=(Act,Loc,\ell^{0},X,T,Inv,\mathrm{AP},V)$ be a DTA, and $v^{0}$ a clock valuation such that ∀x∈X, $v^{0}\left(x\right)=0$.*

*The concrete model for A is a tuple*


$$M_{A}=(Q,q^{0},\overset{}{⟶},V)\mathrm{where},$$


*$Q=Loc\times N^{X}$ is the set of the concrete states.*

*$q^{0}=\left(\ell^{0},v^{0}\right)$ is the initial state.*

*A valuation function $V:Q\to 2^{\mathrm{AP}}$ is defined as V((ℓ,v))=V(ℓ) for each state (ℓ,v)∈Q$\overset{}{⟶}\subseteq Q\times \left(Act\cup N_{+}\right)\times Q$ is a transition relation on Q defined by action and time transitions as follows.*

*For a∈Act and $\delta \in N_{+}$:*
*1*.
*Action transition: $\left(\ell ,v\right)\overset{a}{⟶}\left(\ell^{\prime },v^{\prime }\right)$ if there is a transition $\ell \overset{a,\mathrm{cc},X}{⟶}\ell^{\prime }\in T$ such that v⊨cc∧Inv(ℓ) and $v^{\prime }=v\left[X:=0\right]$ and $v^{\prime }⊨Inv\left(\ell^{\prime }\right)$,*
*2*.
*Time transition: $\left(\ell ,v\right)\overset{\delta }{⟶}\left(\ell ,v+\delta \right)$ iff v⊨Inv(ℓ) and $\left(\forall 0<\delta^{\prime }\leq \delta \right)v+\delta ⊨Inv\left(\ell \right)$.*



Let us observe that for the considered set of clock constraints C(X), the condition of the time transition $\left(l,\;v\right)\;\overset{\delta }{⟶}\;\left(l,\;v+\delta \right)$ can be replaced by a simpler one. Namely, v⊨Inv(ℓ) and v+δ⊨Inv(ℓ).

A *path* ρ in A is an infinite sequence of concrete states $q_{0},q_{1},q_{2},\ldots$ such that for all j∈N, $q_{j}\overset{\mu_{j}}{⟶}q_{j+1}$ for some $\mu_{j}\in N_{+}\cup Act$. Such a definition of the path permits two consecutive actions to be performed one after the other, i.e.,  no time has to elapse between two consecutive actions. It means that we are dealing with the point-based *weakly monotonic* integer-time semantics.

From now on, for a path $\rho =q_{0},q_{1},q_{2},\ldots$, by ρ(m), we denote the state $q_{m}$.

### 2.4. MTL Logic

MTL [14,15] (metric LTL) is the extension of LTL in which temporal operators are replaced by its time-constrained versions. MTL can express many time constraints. For example, we can express a system property: if a system is in the state *q*, then it will be in the state $q^{\prime }$ exactly 3 time units later.

#### 2.4.1. Syntax

We begin with some preliminary definitions. Let p∈AP be an atomic proposition and I the set of all the intervals in N of the form [a,b) or [a,∞), where a,b∈N and a<b, and let I∈I. Observe that we do not exclude one-element intervals since [a,a] can be expressed as [a,a+1). The MTL logic in positive normal form is defined in the following way:$$\alpha \;:=\;\mathrm{true}|\mathrm{false}|p|\neg p|\alpha \wedge \alpha |\alpha \vee \alpha |\alpha U_{I}\alpha |G_{I}\alpha .$$

The operators $U_{I}$ and $G_{I}$ are called *bounded until* and *bounded always*, respectively, and they are read as “until in the interval I” and “always in the interval I”. The operator $F_{I}$ is defined in the standard way: $F_{I}\alpha \overset{\mathrm{df}}{\;=\;}\mathrm{true}\;U_{I}\;\alpha$.

#### 2.4.2. Semantics

There are two possible semantics for metric temporal logic: the “pointwise” semantics and the “continuous” semantics [15]. In the pointwise approach, temporal assertions are interpreted only at time points where the action happens in the observed timed behavior of a system. In the continuous one, it is allowed to assert formulae at arbitrary time points between actions as well. In the presented method, we use the *pointwise* semantics.

Let A be a DTA, and $M_{A}$ the concrete model for A. For a path $\rho =q_{0},q_{1},\ldots$, let $\Gamma_{\rho }\left(j\right)=\left\{i\in N|i<j\;\mathrm{and}\;\mathrm{for}\;\mathrm{some}\;\delta_{i}\in N,q_{i}\overset{\delta_{i}}{⟶}q_{i+1}\right\}$, i.e $\Gamma_{\rho }\left(j\right)$ is a set of indices of time transitions. Now, we define a function $\zeta_{\rho }:N\to N$ such that, for all j⩾0, $\zeta_{\rho }\left(j\right)=\sum_{i\in \Gamma_{\rho }\left(j\right)}\delta_{i}$. For all j⩾0, the function $\zeta_{\rho }\left(j\right)$ returns the value of the global time (called “*duration*” in [15]).

Here and in what follows, we use the convention to omit the model from expressions with ⊨ for the sake of brevity.

 **Definition 3.** *Let α and β be* MTL *formulae. The satisfaction relation ${⊨}_{\mathrm{MTL}}$, which defines truth of an * MTL *formula in the concrete model $M_{A}$ along a path ρ starting at position m∈N, is defined inductively:*
*$\left(\rho ,m\right){⊨}_{\mathrm{MTL}}\;\mathrm{true}$,**$\left(\rho ,m\right){⊭}_{\mathrm{MTL}}\;\mathrm{false}$,**$\left(\rho ,m\right){⊨}_{\mathrm{MTL}}\;p\;\mathrm{iff}\;p\in V\left(\rho \left(m\right)\right)$,**$\left(\rho ,m\right){⊨}_{\mathrm{MTL}}\;\neg p\;\mathrm{iff}\;p\notin V\left(\rho \left(m\right)\right)$,**$\left(\rho ,m\right){⊨}_{\mathrm{MTL}}\;\alpha \wedge \beta \;\mathrm{iff}\;\;\left(\rho ,m\right){⊨}_{\mathrm{MTL}}\;\alpha \;\mathrm{and}\;\left(\rho ,m\right){⊨}_{\mathrm{MTL}}\;\beta$,**$\left(\rho ,m\right){⊨}_{\mathrm{MTL}}\;\alpha \vee \beta \;\mathrm{iff}\;\;\left(\rho ,m\right){⊨}_{\mathrm{MTL}}\;\alpha \;\mathrm{or}\;\left(\rho ,m\right){⊨}_{\mathrm{MTL}}\;\beta$,*$\left(\rho ,m\right){⊨}_{\mathrm{MTL}}\;\alpha U_{I}\beta \;\mathrm{iff}\;\;\left(\exists j\geq m\right)(\zeta_{\rho }\left(j\right)-\zeta_{\rho }\left(m\right)\in I\;\mathrm{and}\;$*$\;\left(\rho ,m+j\right){⊨}_{\mathrm{MTL}}\;\beta \;\mathrm{and}\;\left(\forall m⩽\;j^{\prime }\;<\;j\right)\left(\rho ,m+j^{\prime }\right){⊨}_{\mathrm{MTL}}\;\alpha )$,*$\left(\rho ,m\right){⊨}_{\mathrm{MTL}}\;G_{I}\beta \;\mathrm{iff}\;\left(\forall j\geq m\right)\left(\zeta_{\rho }\left(j\right)-\zeta_{\rho }\left(m\right)\in I\;\mathrm{implies}\left(\rho ,m+j\right){⊨}_{\mathrm{MTL}}\;\beta \right).$

For simplicity of notation, we write ρ instead of (ρ,0). Therefore, we shall write $M_{A},\rho {⊨}_{\mathrm{MTL}}\varphi$ for $M_{A},\left(\rho ,0\right){⊨}_{\mathrm{MTL}}\varphi$. An MTL formula φ is *existentially valid* in the model $M_{A}$, which is denoted as $M_{A}{⊨}_{\mathrm{MTL}}E\varphi$, if and only if $M_{A},\rho {⊨}_{\mathrm{MTL}}\varphi$ for some path ρ starting in the initial state of $M_{A}$. Determining whether an MTL formula φ is existentially valid in a given model is called the *existential model-checking problem*.

## 3. Bounded Model Checking

The verification method presented in this paper is based on the translation of MTL formula to ${\mathrm{LTL}}_{q}$ formula. We extend a standard LTL logic by adding an extra set of propositional variables. We compare our new method with the corresponding method presented in [26].

### 3.1. The Translation


The set of all the clock valuations is infinite, which means that the model has an infinite set of states. We need to abstract the proposed model before we can apply the BMC technique.

#### 3.1.1. Abstract Model

Let $A=(Act,Loc,\ell^{0},T,X,$Inv,AP,V) be a discrete timed automaton with $X=\{x_{0},\ldots ,x_{n-1}\}$. For each j∈{0,…,n−1}, let $c_{j}^{max}$ be the largest constant appearing in any clock constraint involving clock $x_{j}$ and used in the state invariants and guards of A. Two clock valuations *v* and $v^{\prime }$ in $N^{X}$ are equivalent, which is denoted by $v≃v^{\prime }$, if and only if for each 0⩽j<n either $v\left(x_{j}\right)>c_{j}^{max}$ and $v^{\prime }\left(x_{j}\right)>c_{j}^{max}$ or $v\left(x\right)⩽c_{j}^{max}$ and $v^{\prime }\left(x\right)⩽c_{j}^{max}$ and $v\left(x\right)=v^{\prime }\left(x\right)$.

It is easy to see that the relation ≃ is an equivalence relation, which enables us to construct a finite abstract model.

To this end, we define the set of possible values of clock $x_{j}$ in the abstract model as ${I\;D}_{j}=\left\{0,\ldots ,c_{j}^{max}+1\right\}$ for 0⩽j<n. Moreover, for two clock valuations *v* and $v^{\prime }$ in ${I\;D}_{0}\times \ldots \times {I\;D}_{n-1}$, we say that $v^{\prime }$ is the *time successor* of *v* (denoted succ(v)) as follows: for each 0⩽j<n,



$$\begin{array}{c} succ\left(v\right)\left(x_{j}\right)=\left\{\begin{array}{cc} v(x_{j})+1, & \mathrm{if}\;v\left(x_{j}\right)⩽c_{j}^{max},\\ c_{j}^{max}+1, & \mathrm{if}\;v\left(x_{j}\right)=c_{j}^{max}+1. \end{array}\right. \end{array}$$



 **Definition 4.** 
*Let $A=(Act,Loc,\ell^{0},X,T,Inv,\mathrm{AP},V)$ be a discrete timed automaton. The abstract model for A is a tuple*


$$\hat{M}=(\hat{S},s^{0},\overset{}{↪},\hat{V})\mathrm{where},$$


*$\hat{S}=L\times \left({I\;D}_{0}\times \ldots \times {I\;D}_{n-1}\right)$ is the set of abstract states;*

*$s^{0}=\left(\ell^{0},{\left\{0\right\}}^{n}\right)$ is the initial state;*

*$\hat{V}:\hat{S}\to 2^{\mathrm{AP}}$ is a valuation function such that for all p∈AP, $p\in \hat{V}\left(\left(\ell ,v\right)\right)$ if and only if p∈V(ℓ);*

*$\overset{}{↪}\subseteq S\times Act^{\prime }\times S$, where $Act^{\prime }=Act\cup \left\{\tau \right\}$ is a transition relation defined by the time and action transitions.*
−
*The time transition is defined as $\left(\ell ,v\right)\overset{\tau }{↪}\left(\ell ,v^{\prime }\right)$ if and only if v⊨Inv(ℓ), $v^{\prime }=succ\left(v\right)$ and $v^{\prime }⊨Inv\left(\ell \right)$.*
−
*The action transition is defined as follows: for any a∈Act, $\left(\ell ,v\right)\overset{a}{↪}\left(\ell^{\prime },v^{\prime }\right)$ if and only if there exists a transition $\ell \overset{a,\mathrm{cc},X}{⟶}\ell^{\prime }\in T$ such that v⊨cc∧Inv(ℓ), $v^{\prime }=v\left[X:=0\right]$ and $v^{\prime }⊨Inv\left(\ell^{\prime }\right)$.*




 **Definition 5.** *A*path*in the abstract model $\hat{M}$ is a sequence $\pi =(s_{0},s_{1},\ldots )$ of states such that for each j∈N, either $\left(s_{j}\overset{\tau }{↪}s_{j+1}\right)$ or $\left(s_{j}\overset{a}{↪}s_{j+1}\right)$, for some action a∈Act.*

For a given path π, π(j) denotes the *j*-th state $s_{j}$ of the path π, π[..j]=(π(0),…,π(j)) denotes the *j*-th prefix of the path π ending with π(j). Given a path π one can define a function $\zeta_{\pi }:N\mapsto N$ such that, for each j⩾0, $\zeta_{\pi }\left(j\right)$ is equal to the number of time transitions on the prefix π[..j].

 **Definition 6.** *Let $M_{A}$ be the concrete model for A and $\hat{M}$ the abstract model for A. We say that a state q=(l,v) in the concrete model $M_{A}$, and a state $s=(l^{\prime },v^{\prime })$ in the abstract model $\hat{M}$ are*equivalent*, which is denoted by q≅s, if and only if $l=l^{\prime }$ and $v≃v^{\prime }$.*

It is well-known [52] that the relation ≅ is *weak-time-bisimulation equivalent* between the concrete model and the abstract model. The reason is that one can replace one δ-value time transition in the concrete model by δ individual transitions in the abstract model, whereas δ individual transitions in the abstract model can be replaced by one δ-value time transition in the concrete model.

#### 3.1.2. MTL Semantics in the Abstract Model

 **Definition 7.** *The satisfiability relation ${⊨}_{\mathrm{MTL}}^{d}$, which defines the truth of an *MTL* formula in the abstract model $\hat{M}$ along the abstract path π with the starting point m at the depth d⩾m, is inductively defined as follows:**$\left(\pi ,m\right){⊨}_{\mathrm{MTL}}^{d}\mathrm{true}$,**$\left(\pi ,m\right){⊭}_{\mathrm{MTL}}^{d}\mathrm{false}$,**$\left(\pi ,m\right){⊨}_{\mathrm{MTL}}^{d}p$ iff $p\in \hat{V}\left(\pi \left(d\right)\right)$,**$\left(\pi ,m\right){⊨}_{\mathrm{MTL}}^{d}\neg p$ iff $p\notin \hat{V}\left(\pi \left(d\right)\right)$,**$\left(\pi ,m\right){⊨}_{\mathrm{MTL}}^{d}\alpha \wedge \beta \;\mathrm{iff}\;\;\left(\pi ,m\right){⊨}_{\mathrm{MTL}}^{d}\alpha$ and $\left(\pi ,m\right){⊨}_{\mathrm{MTL}}^{d}\beta$,**$\left(\pi ,m\right){⊨}_{\mathrm{MTL}}^{d}\alpha \vee \beta \;\mathrm{iff}\;\;\left(\pi ,m\right){⊨}_{\mathrm{MTL}}^{d}\alpha$ or $\left(\pi ,m\right){⊨}_{\mathrm{MTL}}^{d}\beta$,*$\left(\pi ,m\right){⊨}_{\mathrm{MTL}}^{d}\alpha U_{I}\beta \;\;\mathrm{iff}\;\left(\exists j\geq d\right)(\zeta_{\pi }\left(j\right)-\zeta_{\pi }\left(d\right)\;\in \;I\;\mathrm{and}\;\left(\pi ,d\right){⊨}_{\mathrm{MTL}}^{j}\beta$*$\;\mathrm{and}\;\left(\forall d⩽\;i\;<\;j\right)\left(\pi ,d\right){⊨}_{\mathrm{MTL}}^{i}\alpha )$,**$\left(\pi ,m\right){⊨}_{\mathrm{MTL}}^{d}G_{I}\beta$ iff $\left(\forall j\geq d\right)(\zeta_{\pi }\left(j\right)-\zeta_{\pi }\left(d\right)\;\in \;I$ implies $\left(\pi ,d\right)\;{⊨}_{\mathrm{MTL}}^{j}\;\beta )$.*

In the above definition, *m* does not play itself a part in the satisfaction relation. However, this notation is helpful for Definition 8.

 **Theorem 1** (The equivalence of the MTL semantics in the concrete and abstract models). *Let $\hat{M}$ be the abstract model for the discrete timed automaton A and $M_{A}$ the concrete model for A. Then, for each *MTL* formula φ, the following equivalence holds: $\hat{M},\;\left(\pi ,m\right){⊨}_{\mathrm{MTL}}^{d}\varphi \Leftrightarrow M_{A},\left(\rho ,m\right)\;{⊨}_{\mathrm{MTL}}\;\varphi$.*

 **Proof.** The proof of Theorem 1 follows from the definition of the satisfiability relation and the weak-timed-bisimulation equivalence of the models $M_{A}$ and $\hat{M}$.    □

#### 3.1.3. ${\mathrm{LTL}}_{q}$ Logic

Let I be the set of all intervals and ${\mathrm{AP}}_{I}=\left\{q_{I}|I\in I\right\}$ a set of the new propositional variables. An ${\mathrm{LTL}}_{q}$ formula in the negation normal form is defined by the following grammar:

$$\psi ::=\mathrm{true}|\mathrm{false}|p|\neg p|q_{I}|\neg q_{I}|\psi \wedge \psi |\psi \vee \psi |\psi U\psi |G\psi ,$$
where p∈AP and $q_{I}\in {\mathrm{AP}}_{I}$.


 **Definition 8.** 
*The satisfaction relation ${⊨}^{d}$, which defines the truth of an ${\mathrm{LTL}}_{q}$ formula in the abstract model $\hat{M}$ along the abstract path π at the position m, at depth d⩾m is inductively defined as follows:*

*$\left(\pi ,m\right){⊨}^{d}\mathrm{true}$,*

*$\left(\pi ,m\right)\;{⊭}^{d}\;\mathrm{false}$,*

*$\left(\pi ,m\right){⊨}^{d}p$ iff $p\in \hat{V}\left(\pi \left(d\right)\right)$,*

*$\left(\pi ,m\right){⊨}^{d}\neg p$ iff $p\notin \hat{V}\left(\pi \left(d\right)\right)$,*

*$\left(\pi ,m\right){⊨}^{d}q_{I}$ iff $\zeta_{\pi }\left(d\right)-\zeta_{\pi }\left(m\right)\in I$,*

*$\left(\pi ,m\right){⊨}^{d}\neg q_{I}$ iff $\zeta_{\pi }\left(d\right)-\zeta_{\pi }\left(m\right)\notin I$,*

*$\left(\pi ,m\right){⊨}^{d}\alpha \wedge \beta$ iff $\left(\pi ,m\right){⊨}^{d}\alpha$ and $\left(\pi ,m\right){⊨}^{d}\beta$,*

*$\left(\pi ,m\right){⊨}^{d}\alpha \vee \beta$ iff $\left(\pi ,m\right){⊨}^{d}\alpha$ or $\left(\pi ,m\right){⊨}^{d}\beta$,*

*$\left(\pi ,m\right){⊨}^{d}\alpha U\beta$ iff $\left(\exists j⩾d\right)(\left(\pi ,d\right)\;{⊨}^{j}\;\beta$ and $\left(\forall d⩽i<j\right)\left(\pi ,d\right)\;{⊨}^{i}\;\alpha ))$,*

*$\left(\pi ,m\right){⊨}^{d}G\beta$ iff $\left(\forall j⩾d\right)(\left(\pi ,d\right){⊨}^{j}\beta )$.*



An ${\mathrm{LTL}}_{q}$ formula ψ is *existentially valid* in the abstract model $\hat{M}$, denoted as $\hat{M}\;⊨\;E\psi$, if and only if $\hat{M},\left(\pi ,0\right){⊨}^{0}\psi$ on some path π starting in the initial state of $\hat{M}$.

#### 3.1.4. The Translation from MTL to ${\mathrm{LTL}}_{q}$

Two translations from MTL to LTL were described in [45]. However, in the first translation, the new propositional variables encode time differences between states, and in the second translation a new propositional variable called gap encodes the jumps between states. In the translation presented below, we use *global time* approach [1]. However, in [1] the strongly monotonic semantics was used.

 **Definition 9.** *Let p∈AP, and α, β a* MTL *formulae. The translation from* MTL *to ${\mathrm{LTL}}_{q}$ is defined as a function $\mathrm{tr}:\mathrm{MTL}\to {\mathrm{LTL}}_{q}$ by the following equations:*
*tr(true)=true,**tr(false)=false,**tr(p)=p,**tr(¬p)=¬p,**tr(α∧β)=tr(α)∧tr(β),**tr(α∨β)=tr(α)∨tr(β),**$\mathrm{tr}(\alpha U_{I}\beta )=\mathrm{tr}(\alpha )U\left(q_{I}\wedge \mathrm{tr}(\beta )\right)$, and**$\mathrm{tr}(G_{I}\beta )$ = $G(\neg q_{I}\vee \mathrm{tr}(\beta ))$.*

The translation of the $F_{I}$ operator follows from its definition in terms of the $U_{I}$ operator. Observe that the translation of literals, as well as logical connectives, is straightforward. The translation of the operator $U_{I}$ ensures that the formula β holds at some point in the interval I (it is expressed by the requirement $q_{I}\wedge \mathrm{tr}(\beta )$) and α holds everywhere before β holds. Similarly, the translation of the $G_{I}$ operator ensures that β holds at every point in the interval I (it is expressed by the requirement $\neg q_{I}\vee \mathrm{tr}(\beta )$).

The translation from EMTL to E${\mathrm{LTL}}_{q}$ is more straightforward than the one presented in [48], e.g., TPTL expressiveness is higher than ${\mathrm{LTL}}_{q}$. In our case, we do not need this extension of the logic to solve the given problem.

 **Theorem 2.** *Let A be a discrete timed automaton, φ an *MTL* formula, and $\hat{M}$ the abstract model for A. Then $\hat{M}\;{⊨}_{\mathrm{MTL}}\;E\;\varphi$ if, and only if $\hat{M}\;⊨\;E\;\mathrm{tr}(\varphi )$.*

## 4. Proof of the Theorem 2

A proof of the Theorem 2 follows directly from the Lemmas 1 and 2.

 **Lemma 1.** *Let A be a discrete timed automaton, φ an *MTL* formula, $\hat{M}$ an abstract model for discrete timed automaton A, and π an abstract path in the abstract model $\hat{M}$. If $\left(\pi ,m\right)\;{⊨}_{\mathrm{MTL}}^{d}\;\varphi$, then $\left(\pi ,m\right){⊨}^{d}\mathrm{tr}(\varphi )$.*

 **Proof.** We proceed by induction on the length of a given formula.

Assume that $\left(\pi ,m\right)\;{⊨}_{\mathrm{MTL}}^{d}\;\varphi$. Consider the following cases:
φ∈AP. Because tr(φ)=φ, it is obvious that tr(φ)∈AP. Therefore, $\left(\pi ,m\right)\;{⊨}_{\mathrm{MTL}}^{d}\;\varphi$$\Leftrightarrow \varphi \in \hat{V}\left(\pi \left(d\right)\right)\Leftrightarrow \mathrm{tr}(\varphi )\in \hat{V}\left(\pi \left(d\right)\right)\Leftrightarrow \left(\pi ,m\right){⊨}^{d}\mathrm{tr}(\varphi )$.φ=¬p, where p∈AP. Thus, tr(φ)=φ. Therefore, $\left(\pi ,m\right)\;{⊨}_{\mathrm{MTL}}^{d}\varphi \Leftrightarrow \varphi \notin V\left(\pi \left(d\right)\right)\Leftrightarrow \left(\pi ,m\right)\;{⊨}_{\mathrm{MTL}}\neg p\Leftrightarrow \left(\pi ,m\right)\;{⊨}^{d}\varphi \Leftrightarrow \left(\pi ,m\right)\;{⊨}^{d}\mathrm{tr}(\varphi )$.φ=α∧β. From the definition of the satisfiability relation (Definition 7) it follows that $\left(\pi ,m\right)\;{⊨}_{\mathrm{MTL}}^{d}\;\alpha$ and $\left(\pi ,m\right)\;{⊨}_{\mathrm{MTL}}^{d}\;\beta$. By inductive hypothesis, we obtain $\left(\pi ,m\right){⊨}^{d}\mathrm{tr}(\alpha )$ and $\left(\pi ,m\right){⊨}^{d}\mathrm{tr}(\beta )$. Therefore, $\left(\pi ,m\right){⊨}^{d}\mathrm{tr}(\alpha )\wedge \mathrm{tr}(\beta )$, and hence $\left(\pi ,m\right){⊨}^{d}\mathrm{tr}(\alpha \wedge \beta )\Leftrightarrow \left(\pi ,m\right){⊨}^{d}\mathrm{tr}(\varphi )$.φ=α∨β. From the definition of the satisfiability relation (Definition 7) it follows that $\left(\pi ,m\right)\;{⊨}_{\mathrm{MTL}}^{d}\;\alpha$ or $\left(\pi ,m\right)\;{⊨}_{\mathrm{MTL}}^{d}\;\beta$. By inductive hypothesis, we obtain that $\left(\pi ,m\right){⊨}^{d}\mathrm{tr}(\alpha )$ or $\left(\pi ,m\right){⊨}^{d}\mathrm{tr}(\beta )$. Therefore, $\left(\pi ,m\right){⊨}^{d}\mathrm{tr}(\alpha )\vee \mathrm{tr}(\beta )$, and hence $\left(\pi ,m\right){⊨}^{d}\mathrm{tr}(\alpha \vee \beta )\Leftrightarrow \left(\pi ,m\right){⊨}^{d}\mathrm{tr}(\varphi )$.$\varphi =\alpha U_{I}\beta$. Assume that $\left(\pi ,m\right)\;{⊨}_{\mathrm{MTL}}^{d}\;\varphi$. From the definition of the satisfiability relation (Definition 7), it follows that $\zeta_{\pi }\left(j\right)-\zeta_{\pi }\left(d\right)\;\in \;I$ and $(\left(\pi ,d\right)\;{⊨}_{\mathrm{MTL}}^{j}\;\beta$ and $\left(\forall d⩽\;i\;<\;j\right)\left(\pi ,d\right)\;{⊨}_{\mathrm{MTL}}^{i}\;\alpha )$, for some j≥d. By inductive hypothesis, we obtain $\zeta_{\pi }\left(j\right)-\zeta_{\pi }\left(d\right)\;\in \;I$ and $\left(\pi ,d\right){⊨}^{j}\mathrm{tr}(\beta )$, for some j≥d and $\left(\pi ,d\right){⊨}^{i}\mathrm{tr}(\alpha )$, for all d⩽i<j. Therefore, $\left(\pi ,d\right)\;{⊨}^{j}\;q_{I}\wedge \mathrm{tr}(\beta )$, for some j≥d, and $\left(\pi ,d\right)\;{⊨}^{i}\;\mathrm{tr}(\alpha )$, for all d⩽i<j. Therefore, we conclude that $\left(\pi ,m\right){⊨}^{d}\mathrm{tr}(\alpha U_{I}\beta )$.$\varphi =G_{I}\beta$. Assume that $\left(\pi ,m\right)\;{⊨}_{\mathrm{MTL}}^{d}\;\varphi$. From the definition of the satisfiability relation (Definition 7), it follows that $\left(\forall j\geq d\right)(\zeta_{\pi }\left(j\right)-\zeta_{\pi }\left(d\right)\;\in \;I$ implies $\left(\pi ,d\right)\;{⊨}_{\mathrm{MTL}}^{j}\;\beta )$, which means that $\zeta_{\pi }\left(j\right)-\zeta_{\pi }\left(d\right)\;\notin \;I\vee \left(\pi ,d\right)\;{⊨}_{\mathrm{MTL}}^{j}\;\beta$, for all j≥d. By inductive hypothesis, we obtain $\zeta_{\pi }\left(j\right)-\zeta_{\pi }\left(d\right)\;\notin \;I\vee \left(\pi ,d\right){⊨}^{j}\mathrm{tr}(\beta )$, for all j≥d. Hence, $\left(\pi ,d\right)\;{⊨}^{j}\;\neg q_{I}\wedge \mathrm{tr}(\beta )$, for all j≥d. From the semantics of ${\mathrm{LTL}}_{q}$, it follows that $\left(\pi ,m\right){⊨}^{d}G\left(\neg q_{I}\vee \mathrm{tr}(\beta )\right)$. So, we can conclude that $\left(\pi ,m\right){⊨}^{d}\mathrm{tr}(G_{I}\beta )$.
□

 **Lemma 2.** *Let A be a discrete timed automaton, φ an* MTL *formula, $\hat{M}$ an abstract model for the discrete timed automaton A, and π an abstract path in the abstract model $\hat{M}$. If $\left(\pi ,m\right)\;{⊨}^{d}\mathrm{tr}(\varphi )$, then $\left(\pi ,m\right)\;{⊨}_{\mathrm{MTL}}^{d}\;\varphi$.*

 **Proof.** We proceed by induction on the length of a given formula.
φ∈AP. Since φ=tr(φ), it follows that φ∈AP. Therefore, $\left(\pi ,m\right)\;{⊨}^{d}\;\mathrm{tr}(\varphi )\Leftrightarrow \mathrm{tr}(\varphi )\in \hat{V}\left(\pi \left(d\right)\right)\Leftrightarrow \varphi \in \hat{V}\left(\pi \left(d\right)\right)\Leftrightarrow \left(\pi ,m\right){⊨}_{\mathrm{MTL}}^{d}\varphi$.φ=¬p, where p∈AP. Then φ=tr(φ). Therefore, $\left(\pi ,m\right)\;{⊨}^{d}\mathrm{tr}(\varphi )\Leftrightarrow \mathrm{tr}(\varphi )\notin V\left(\pi \left(d\right)\right)\Leftrightarrow \varphi \notin V\left(\pi \left(d\right)\right)\Leftrightarrow \left(\pi ,m\right)\;{⊨}_{\mathrm{MTL}}^{d}\neg p\Leftrightarrow \left(\pi ,m\right)\;{⊨}_{\mathrm{MTL}}^{d}\varphi$.φ=α∧β. Thus, tr(φ)=tr(α∧β)=tr(α)∧tr(β). From the definition of the satisfiability relation (Definiton 8) it follows that $\left(\pi ,m\right)\;{⊨}^{d}\;\mathrm{tr}(\alpha )$ and $\left(\pi ,m\right)\;{⊨}^{d}\;\mathrm{tr}(\beta )$. By inductive hypothesis, we obtain $\left(\pi ,m\right){⊨}_{\mathrm{MTL}}^{d}\alpha$ and $\left(\pi ,m\right){⊨}_{\mathrm{MTL}}^{d}\beta$. Hence, $\left(\pi ,m\right){⊨}_{\mathrm{MTL}}^{d}\alpha \wedge \beta$ and thus $\left(\pi ,m\right){⊨}_{\mathrm{MTL}}^{d}\alpha \wedge \beta \Leftrightarrow \left(\pi ,m\right){⊨}_{\mathrm{MTL}}^{d}\varphi$.φ=α∨β. Then tr(φ)=tr(α∨β)=tr(α)∨tr(β). From the definition of the satisfiability relation (Definition 8) it follows that $\left(\pi ,m\right)\;{⊨}^{d}\;\mathrm{tr}(\alpha )$ or $\left(\pi ,m\right)\;{⊨}^{d}\;\mathrm{tr}(\beta )$. By inductive hypothesis, we obtain $\left(\pi ,m\right){⊨}_{\mathrm{MTL}}^{d}\mathrm{tr}(\alpha )$ or $\left(\pi ,m\right){⊨}_{\mathrm{MTL}}^{d}\mathrm{tr}(\beta )$. Hence, $\left(\pi ,m\right){⊨}^{d}\alpha \vee \beta$, and thus $\left(\pi ,m\right){⊨}_{\mathrm{MTL}}^{d}\alpha \vee \beta \Leftrightarrow \left(\pi ,m\right){⊨}_{\mathrm{MTL}}^{d}\varphi$.$\varphi =\alpha U_{I}\beta$. Assume that $\left(\pi ,m\right)\;{⊨}_{\mathrm{MTL}}^{d}\;\varphi$. From the definition of the translation, it follows that $\left(\pi ,m\right)\;{⊨}^{d}\;\mathrm{tr}(\alpha )U\left(q_{I}\wedge \mathrm{tr}(\beta )\right)$. From the definition of the satisfiability relation 8, it follows that $\left(\pi ,d\right)\;{⊨}^{j}\;q_{I}\wedge \mathrm{tr}(\beta )$ and $\left(\forall d⩽i<j\right)\left(\pi ,d\right)\;{⊨}^{i}\;\mathrm{tr}(\alpha )$, for some j⩾d. Therefore, $\left(\pi ,d\right)\;{⊨}^{j}\;\zeta_{\pi }\left(j\right)-\zeta_{\pi }\left(d\right)\;\in \;I\wedge \left(\pi ,d\right)\;{⊨}^{j}\;\mathrm{tr}(\beta )$ and $\left(\forall d⩽i<j\right)\left(\pi ,d\right)\;{⊨}^{i}\;\mathrm{tr}(\alpha )$, for some j⩾d. From the inductive hypothesis, we obtain $\left(\pi ,d\right)\;{⊨}_{\mathrm{MTL}}^{j}\;\zeta_{\pi }\left(j\right)-\zeta_{\pi }\left(d\right)\;\in \;I\wedge \left(\pi ,d\right)\;{{⊨}_{\mathrm{MTL}}}^{j}\;\beta$ and $\left(\forall d⩽i<j\right)\left(\pi ,d\right)\;{⊨}_{\mathrm{MTL}}^{i}\;\alpha$, thus $\left(\pi ,d\right)\;{⊨}_{\mathrm{MTL}}^{j}\;\zeta_{\pi }\left(j\right)-\zeta_{\pi }\left(d\right)\;\in \;I\wedge \beta$ and $\left(\forall d⩽i<j\right)\left(\pi ,d\right)\;{⊨}_{\mathrm{MTL}}^{i}\;\alpha$. Thus, we conclude that $\left(\pi ,m\right)\;{⊨}_{\mathrm{MTL}}^{d}\alpha U_{I}\beta$.$\varphi =G_{I}\beta$. Assume that $\left(\pi ,m\right)\;{⊨}^{d}\;\varphi$. From the definition of the translation, it follows that $\left(\pi ,m\right)\;{⊨}^{d}\;G\left(\neg q_{I}\vee \mathrm{tr}(\beta )\right)$. From the definition of the satisfiability relation $\left(\forall j⩾d\right)(\left(\pi ,d\right){⊨}^{j}\neg q_{I}$ or $\left(\pi ,d\right){⊨}^{j}\mathrm{tr}(\beta ))$, which means $(\left(\pi ,d\right){⊨}^{j}\zeta_{\pi }\left(j\right)-\zeta_{\pi }\left(d\right)\;\notin \;I$ or $\left(\pi ,d\right){⊨}^{j}\mathrm{tr}(\beta ))$, for all j⩾d. By inductive hypothesis, we obtain $\left(\forall j⩾d\right)(\left(\pi ,d\right)\;{⊨}_{\mathrm{MTL}}^{j}\;\zeta_{\pi }\left(j\right)-\zeta_{\pi }\left(d\right)\;\notin \;I$ or $\left(\pi ,d\right){⊨}^{j}\beta )$, which is equivalent to $\left(\forall j⩾d\right)(\left(\pi ,d\right)\;{⊨}_{\mathrm{MTL}}^{j}\;\zeta_{\pi }\left(j\right)-\zeta_{\pi }\left(d\right)\;\notin \;I\vee \beta )$. Therefore, $\left(\pi ,m\right)\;{{⊨}_{\mathrm{MTL}}}^{d}\;G_{I}\beta$.
□

 **Proof** **of Theorem 2.** (⟹) Assume that $\hat{M}\;{⊨}_{\mathrm{MTL}}\;E\varphi$. Therefore, $\hat{M},\pi \;{⊨}_{\mathrm{MTL}}^{0}\varphi$, for some abstract path π in $\hat{M}$ such that $\pi \left(0\right)=s_{0}$. It means that $\hat{M},\left(\pi ,0\right)\;{⊨}_{\mathrm{MTL}}^{0}\varphi$. From Lemma 1, it follows that $\left(\pi ,0\right){⊨}^{0}\mathrm{tr}(\varphi )$. Therefore, $\left(\pi ,m\right){⊨}^{0}\mathrm{tr}(\varphi )$, for m=0. Thus $\hat{M}\;⊨\;E\mathrm{tr}(\varphi )$.(⟸) Assume that $\hat{M}\;⊨\;E\mathrm{tr}(\varphi )$. Hence, $\hat{M},\pi {⊨}^{0}\;\mathrm{tr}(\varphi )$, for some abstract path π in $\hat{M}$ such that $\pi \left(0\right)=s_{0}$. It means that $\left(\pi ,0\right)\;{⊨}^{0}\;\mathrm{tr}(\varphi )$. From Lemma 2, it follows that $\hat{M},\left(\pi ,0\right){⊨}_{\mathrm{MTL}}^{0}\varphi$. Therefore, $\hat{M},\left(\pi ,m\right){⊨}_{\mathrm{MTL}}^{0}\varphi$, for m=0. Thus $\hat{M}\;{⊨}_{\mathrm{MTL}}\;E\varphi$.    □

### 4.1. Bounded Semantics

To define the bounded semantics, we need to represent infinite paths in the abstract model using *k*-paths and loops [20,21].

 **Definition 10.** 
*Let $\hat{M}$ be an abstract model, k∈N and 0⩽l⩽k. A k-path is a pair (π,l), which is also denoted as $\pi_{l}$, where π is a finite sequence of the abstract states $\pi =(s_{0},\ldots ,s_{k})$ such that for each 0⩽j<k, either $\left(s_{j}\overset{\tau }{↪}s_{j+1}\right)$ or $\left(s_{j}\overset{a}{↪}s_{j+1}\right)$, for some a∈Act. Moreover, every action transition is preceded by at least one time transition. A k-path $\pi_{l}$ is a loop, written as $⅁(\pi_{l})$ for short, if l<k and π(k)=π(l).*


If a *k*-path $\pi_{l}$ is a loop, it represents the infinite path of the form $uv^{\omega }$, where u=(π(0),…,π(l)) and v=(π(l+1),…,π(k)). We denote this unique path by $\tilde{\pi_{l}}$. Note that for each j∈N, $\tilde{\pi_{l}}\left(l+j\right)=\tilde{\pi_{l}}\left(k+j\right)$.

Given a path $\tilde{\pi_{l}}$, one can define a function $\zeta_{\tilde{\pi_{l}}}:N\mapsto N$ such that for each j⩾0, $\zeta_{\tilde{\pi_{l}}}\left(j\right)$ is equal to the number of time transitions on the prefix $\tilde{\pi_{l}}\left[..j\right]$. Note that for each j⩾0, $\zeta_{\tilde{\pi_{l}}}\left(j\right)$ gives the value of the global time in the *j*-th state.

In the definition of bounded semantics for variables from ${\mathrm{AP}}_{I}$, one needs to use only a finite prefix of the sequence $\left(\zeta_{\tilde{\pi_{l}}}\left(0\right),\zeta_{\tilde{\pi_{l}}}\left(1\right),\ldots \right)$. Namely, for a *k*-path $\pi_{l}$ that is not a loop, the prefix of the length *k* is needed, and for a *k*-path $\pi_{l}$ that is a loop, the prefix of the length k+k−l is needed.

 **Definition 11** (Bounded semantics). *Let $\hat{M}$ be the abstract model, $\pi_{l}$ a k-path in $\hat{M}$, 0⩽m⩽k, and 0⩽d⩽k. The relation ${⊨}_{k}^{d}$ is defined inductively as follows:*
$$\begin{array}{lll} \left(\pi_{l},m\right)\;{⊨}_{k}^{d}\;\mathrm{true}, & & \\ \left(\pi_{l},m\right)\;{⊭}_{k}^{d}\;\mathrm{false}, & & \\ \left(\pi_{l},m\right)\;{⊨}_{k}^{d}\;p & \mathrm{iff} & p\in V(\pi_{l}\left(d\right)),\\ \left(\pi_{l},m\right)\;{⊨}_{k}^{d}\;\neg p & \mathrm{iff} & p\notin V(\pi_{l}\left(d\right))\\ \left(\pi_{l},m\right)\;{⊨}_{k}^{d}\;q_{I} & \mathrm{iff} & \left\{\begin{array}{cc} \zeta_{\pi_{l}}\left(d\right)-\zeta_{\pi_{l}}\left(m\right)\in I, & \mathrm{if}\;\neg ⅁(\pi_{l}),\\ \zeta_{\pi_{l}}\left(d\right)-\zeta_{\pi_{l}}\left(m\right)\in I, & \mathrm{if}\;⅁(\pi_{l})\;\mathrm{and}\;d⩾m,\\ \zeta_{\tilde{\pi_{l}}}\left(d+k-l\right)-\zeta_{\tilde{\pi_{l}}}\left(m\right)\in I, & \mathrm{if}\;⅁(\pi_{l})\;\mathrm{and}\;d<m, \end{array}\right.\\ \left(\pi_{l},m\right)\;{⊨}_{k}^{d}\;\neg q_{I} & \mathrm{iff} & \left(\pi_{l},m\right)\;\neg {⊭}_{k}^{d}\;q_{I},\\ \left(\pi_{l},m\right)\;{⊨}_{k}^{d}\;\alpha \wedge \beta & \mathrm{iff} & \left(\pi_{l},m\right)\;{⊨}_{k}^{d}\;\alpha \mathrm{and}\left(\pi_{l},m\right)\;{⊨}_{k}^{d}\;\beta ,\\ \left(\pi_{l},m\right)\;{⊨}_{k}^{d}\;\alpha \vee \beta & \mathrm{iff} & \left(\pi_{l},m\right)\;{⊨}_{k}^{d}\;\alpha \mathrm{or}\left(\pi_{l},m\right)\;{⊨}_{k}^{d}\;\beta ,\\ \left(\pi_{l},m\right){⊨}_{k}^{d}\alpha U\beta & \mathrm{iff} & \left(\exists_{d⩽j⩽k}\right)(\left(\pi_{l},d\right){⊨}_{k}^{j}\beta \mathrm{and}(\forall_{d⩽i<j})(\pi_{l},d){⊨}_{k}^{i}\alpha )\mathrm{or}(⅁(\pi_{l})\mathrm{and}(\exists_{l<j<d})(\pi_{l},d){⊨}_{k}^{j}\beta \mathrm{and}(\forall_{l<i<k})\;(\pi_{l},d){⊨}_{k}^{j}\alpha \mathrm{and}(\forall_{d⩽i⩽k})\;(\pi_{l},d){⊨}_{k}^{i}\alpha ),\\ \left(\pi_{l},m\right){⊨}_{k}^{d}G\beta & \mathrm{iff} & \;⅁\left(\pi_{l}\right)\;\mathrm{and}\;(\forall_{j⩽k})j⩾\min(d,l)\mathrm{implies}(\pi_{l},d){⊨}_{k}^{j}\beta . \end{array}$$

The proof of Lemma 3 below is based on induction on the length of the given formula. It is analogous to the proof of Lemma 7 from the paper [20].

 **Lemma 3.** 
*Let A be a discrete timed automaton, φ an ${\mathrm{LTL}}_{q}$ formula, and $\hat{M}$ the abstract model for the automaton A. For each ${\mathrm{LTL}}_{q}$ formula φ, each k− path $\pi_{l}$ in $\hat{M}$, each 0≤m≤k and each 0≤d≤k, if  $\left(\pi_{l},m\right){⊨}_{k}^{d}\varphi$, there exists a path $\pi^{\prime }$ such that $\pi^{\prime }\left[..k\right]=\pi_{l}$ and m≤d and $\left(\pi^{\prime },m\right){⊨}^{d}\varphi$ or m>d and $\left(\pi^{\prime },m\right){⊨}^{d+k-l}\varphi$.*


The proof of the Lemma 4 below is based on the well-known fact that if the LTL formula is true on some infinite path, it is also true on an infinite path of the form $uv^{\omega }$, where *u* and *v* are finite sequences of states [20].

 **Lemma 4.** 
*Let A be a discrete timed automaton, φ an ${\mathrm{LTL}}_{q}$ formula, $\hat{M}$ the abstract model for the automaton A, π a path in the abstract model, and k≥0. For each ${\mathrm{LTL}}_{q}$ formula φ, each 0≤m≤k and each 0≤d≤k, if  $\left(\pi ,m\right){⊨}^{d}\varphi$, there exists a k− path $\pi_{l}$ such that $\left(\pi_{l},m\right){⊨}_{k}^{d}\varphi$.*


An ${\mathrm{LTL}}_{q}$ formula φ existentially *k*-holds in the model $\hat{M}$, written as $\hat{M}\;{⊨}_{k}\;E\;\varphi$, if and only if $\hat{M},\left(\pi_{l},0\right){⊨}_{k}^{0}\;\varphi$ for some k− path $\pi_{l}$ starting at the initial state.

Theorem 3 shows that for some specific bound, bounded and unbounded semantics are equivalent. The proof of Theorem 3 follows directly from Lemmas 3 and 4.

 **Theorem 3.** 
*Let $\hat{M}$ be the abstract model and φ an ${\mathrm{LTL}}_{q}$ formula. Then, $\hat{M}⊨E\;\varphi$ if and only if there exists a k⩾0 such that $\hat{M}{⊨}_{k}E\;\varphi$.*


 **Example 1.** 
*Figure 1 shows an automaton modeling a simple light switch. It consists of two locations, a and b. When the action on is performed, the clock $x_{0}$ is reset. The automaton can stay in the location b until the valuation of the clock $x_{0}$ is less or equal to 6. The transition from location b to location a (action off) can be performed when the valuation of the clock $x_{0}$ is greater than 3.*

*Figure 2 shows an abstract path in the abstract model for the simple light switch. Under the states, we show the global time at the given position.*
 **Example** **2.** 
*Let us check the satisfiability of formula $q_{I}$ for the case when $\pi_{l}$ is not a loop. Let I=[0,5), k=2, m=3 and d=5.*


$$\left(\pi_{2},3\right){⊨}_{8}^{5}q_{I}\Leftrightarrow \zeta_{\pi }\left(5\right)-\zeta_{\pi }\left(3\right)\in I\Leftrightarrow 2\in I.$$


*Figure 3 shows an example of the k− path, which is a loop. Note that for l=2 and k=8, u=(π(0),π(1),π(2)), and v=(π(3),π(4),π(5),π(6),π(7),π(8)). Under the states, we show the global time in the given state.*


 **Example 3.** 
*Let us check the satisfiability of formula $q_{I}$ for the case when $\pi_{l}$ is a loop and d≥m. Let I=[0,5), k=8, l=2, m=3 and d=5.*


$$\left(\pi_{2},3\right){⊨}_{8}^{5}q_{I}\Leftrightarrow \zeta_{\tilde{\pi_{l}}}\left(5\right)-\zeta_{\tilde{\pi_{l}}}\left(3\right)\in I\Leftrightarrow 2\in I.$$



 **Example 4.** 
*Let us check the satisfiability of the formula $q_{I}$ for the case when $\pi_{l}$ is a loop and d<m. Let I=[0,5), k=8, l=2, m=8 and d=5.*


$$\left(\pi_{2},8\right){⊨}_{8}^{5}q_{I}\Leftrightarrow \zeta_{\tilde{\pi_{l}}}\left(5+8-2\right)-\zeta_{\tilde{\pi_{l}}}\left(8\right)\in I\Leftrightarrow \zeta_{\tilde{\pi_{l}}}\left(11\right)-\zeta_{\tilde{\pi_{l}}}\left(8\right)\in I\Leftrightarrow 4\in I.$$



### 4.2. Translation to SAT

The last step of our method is the standard one ([26,53]). It consists in encoding the transition relation of $\hat{M}$ and the ${\mathrm{LTL}}_{q}$ formula tr(φ). The only novelty lies in the encoding of the finite prefix of the sequence $\left(\zeta_{\tilde{\pi_{l}}}\left(0\right),\zeta_{\tilde{\pi_{l}}}\left(1\right),\ldots \right)$.

Let $\hat{M}$ be the abstract model for the automaton A, tr(φ) be the ${\mathrm{LTL}}_{q}$ formula, and k≥0 a bound. The formula ${\left[\mathrm{tr}(\varphi )\right]}_{k}$ encodes a bounded semantics of the ${\mathrm{LTL}}_{q}$ formula tr(φ). It is defined over the same set of the propositional variables as the propositional formula ${\left[{\hat{M}}^{\mathrm{tr}(\varphi ),s^{0}}\right]}_{k}$.

The definition of the formula ${\left[{\hat{M}}^{\mathrm{tr}(\varphi ),s^{0}}\right]}_{k}$ assumes that, states and actions in the abstract model $\hat{M}$, and passage of time are encoded symbolically. This is possible if the set of states and the set of actions are finite. Formally, each symbolic abstract state $\hat{s}\in \hat{S}$ is represented by a vector, $\bar{w}=\left(\left(w_{1},v_{1}\right),\ldots ,\left(w_{r},v_{r}\right)\right)$ of propositional variables, where the length *r* depends on the number of states in the abstract model. This vector is called *a symbolic state*. Each action a∈Act is represented by a vector $\bar{a}=\left(a_{1},\ldots ,a_{t}\right)$ of propositional variables, where the length *t* depends on number of local actions in A. It is called *a symbolic action*.

A pair consisting of a sequence of the symbolic transitions and a symbolic number is called *a symbolic k-path*. Let π be a pair which represents a symbolic *k*-path: $\left(\left({\bar{w}}_{0},{\bar{w}}_{1},\ldots ,{\bar{w}}_{k-1},{\bar{w}}_{k}\right),\bar{u}\right)$, where ${\bar{w}}_{i}$ is a symbolic state, for 0≤i≤k, and $\bar{u}$ is a symbolic number, which is a vector $\bar{u}=\left(u_{1},\ldots ,u_{y}\right)$ of propositional variables with $y=max(1,\left\lceil \log_{2}\left(k+1\right)\right\rceil )$. Moreover, let ${\bar{a}}_{i}$, for 0<i≤k, be a symbolic action.

Let $\bar{w}$ and ${\bar{w}}^{\prime }$ be two different symbolic states, $\bar{a}$ a symbolic action and $\bar{u}$ a symbolic number. To define the formula ${\left[{\hat{M}}^{\mathrm{tr}(\varphi ),s^{0}}\right]}_{k}$, we use the following auxiliary propositional formulae: $I_{\hat{s}}\left(\bar{w}\right)$ encodes a state $\hat{s}$ in the abstract model $\hat{M}$, $H(\bar{w},{\bar{w}}^{\prime })$ encodes the equality of two global states, $T_{Act}\left(\bar{w},\bar{a},{\bar{w}}^{\prime }\right)$ encodes an action transition in $\hat{M}$, $T_{\tau }\left(\bar{w},\tau ,{\bar{w}}^{\prime }\right)$ encodes a time transition in $\hat{M}$, $B_{j}^{∼}\left(\bar{u}\right)$, for ∼∈{<,≤,=,>,≥} encodes the relation ∼ between *j* and $\bar{u}$, and $L_{k,m}^{l}\left(\pi \right)$ encodes the existence of a loop for path π at position *l*.

The propositional formula ${\left[{\hat{M}}^{\mathrm{tr}(\varphi ),s^{0}}\right]}_{k}$, encodes the unfolding of the transition relation of the abstract model $\hat{M}$ to the depth *k* in the following way:

$${\left[{\hat{M}}^{\mathrm{tr}(\varphi ),s^{0}}\right]}_{k}\;:=\;{⋁}_{s\in s^{0}}I_{s}\left({\bar{w}}_{0}\right)\wedge {⋁}_{l=0}^{k}B_{l}^{=}\left(\bar{u}\right)\wedge ({⋀}_{j=0}^{k-1}\left(T_{\tau }\left({\bar{w}}_{j},\tau ,{\bar{w}}_{j+1}\right)\vee T_{Act}\left({\bar{w}}_{j},{\bar{a}}_{j},{\bar{w}}_{j+1}\right)\right))$$,
where ${\bar{w}}_{i}$, ${\bar{a}}_{i}$ and $\bar{u}$ are, respectively, symbolic states, symbolic actions and the symbolic number for 0≤i≤k.

The next step of the method is the translation of the ${\mathrm{LTL}}_{q}$ formula tr(φ) into the propositional formula ${\left[\mathrm{tr}(\varphi )\right]}_{k}\;:=\;{\left[\mathrm{tr}(\varphi )\right]}_{\left[k,0\right]}^{0}$. To translate the formula tr(φ) to SAT problem, we use the auxiliary propositional formulae defined in [53] and the propositional formula ${Gt}_{I}^{d,m}\left(\pi \right)$. The formula ${Gt}_{I}^{d,m}\left(\pi \right)$ encodes the condition that says that the difference of the symbolic global time at the depth *d* and in the starting point *m* on the symbolic path π belongs to the interval I.

 **Definition 12** (The translation from ${\mathrm{ELTL}}_{q}$ to SAT). *Let $\hat{M}$ be the abstract model, tr(φ) an ${\mathrm{LTL}}_{q}$ formula, and k≥0 a bound. The translation of the formula tr(φ) on the path starting at point m at the depth d is defined inductively:*
$$\begin{array}{cc} {\left[\mathrm{true}\right]}_{\left[k,m\right]}^{d}:=\mathrm{true},\\ {\left[\mathrm{false}\right]}_{\left[k,m\right]}^{d}:=\mathrm{false},\\ {\left[p\right]}_{\left[k,m\right]}^{d}:=p\left({\bar{w}}_{d}\right),\\ {\left[\neg p\right]}_{\left[k,m\right]}^{d}:=\neg p\left({\bar{w}}_{d}\right),\\ {\left[q_{I}\right]}_{\left[k,m\right]}^{d}\;:=\; & \left\{\begin{array}{cc} \overset{k-1}{\underset{l=0}{⋁}}\left({Gt}_{I}^{d,m}\left(\pi \right)\wedge \neg H\left({\bar{w}}_{k},{\bar{w}}_{l}\right)\right)\;\vee & \\ \;\;\;\overset{k-1}{\underset{l=0}{⋁}}\left({Gt}_{I}^{d,m}\left(\pi \right)\wedge H\left({\bar{w}}_{k},{\bar{w}}_{l}\right)\right), & \mathrm{if}\;d\geq m\\ \overset{k-1}{\underset{l=0}{⋁}}\left({Gt}_{I}^{d,m}\left(\pi \right)\wedge \neg H\left({\bar{w}}_{k},{\bar{w}}_{l}\right)\right)\;\vee & \\ \;\;\;\overset{k-1}{\underset{l=0}{⋁}}\left({Gt}_{I}^{d+k-l,m}\left(\pi \right)\wedge H\left({\bar{w}}_{k},{\bar{w}}_{l}\right)\right), & \mathrm{if}\;d<m \end{array}\right.\\ {\left[\neg q_{I}\right]}_{\left[k,m\right]}^{d}\;:=\; & \neg {\left[q_{I}\right]}_{\left[k,m\right]}^{d},\\ {\left[\alpha \wedge \beta \right]}_{\left[k,m\right]}^{d}:=\; & {\left[\alpha \right]}_{\left[k,m\right]}^{d}\wedge {\left[\beta \right]}_{\left[k,m\right]}^{d},\\ {\left[\alpha \vee \beta \right]}_{\left[k,m\right]}^{d}:=\; & {\left[\alpha \right]}_{\left[k,m\right]}^{d}\vee {\left[\beta \right]}_{\left[k,m\right]}^{d},\\ {\left[\alpha U\beta \right]}_{\left[k,m\right]}^{d}\;:=\; & \overset{k}{\underset{j=d}{⋁}}\left({\left[\beta \right]}_{\left[k,m\right]}^{j}\wedge \overset{j-1}{\underset{i=d}{⋀}}{\left[\alpha \right]}_{\left[k,m\right]}^{i}\right)\vee (\overset{d-1}{\underset{l=0}{⋁}}\left(L_{k,m}^{l}\left(\pi \right)\right)\wedge \overset{d-1}{\underset{j=0}{⋁}}(B_{j}^{>}\left(\bar{u}\right)\\ \; & \wedge {\left[\beta \right]}_{\left[k,m\right]}^{j}\wedge \overset{j-1}{\underset{i=0}{⋁}}\left(B_{i}^{>}\left(\bar{u}\right)\to {\left[\alpha \right]}_{\left[k,m\right]}^{i}\right)\wedge \overset{k}{\underset{i=d}{⋀}}{\left[\alpha \right]}_{\left[k,m\right]}^{i})), \end{array}$$
$$\begin{array}{cc} {\left[G\alpha \right]}_{\left[k,m\right]}^{d}\;:=\; & \overset{k-1}{\underset{l=0}{⋁}}\left(L_{k,m}^{l}\left(\pi \right)\right)\wedge \overset{d-1}{\underset{j=0}{⋀}}\left(B_{j}^{⩾}\left(\bar{u}\right)\to {\left[\alpha \right]}_{\left[k,m\right]}^{j}\right)\wedge \overset{k}{\underset{j=d}{⋀}}{\left[\alpha \right]}_{\left[k,m\right]}^{j}. \end{array}$$

 **Theorem 4.** 
*Let $\hat{M}$ be the abstract model. Then for every k∈N, at the depth d≤k, $\hat{M}{⊨}_{k}^{d}E\mathrm{tr}(\varphi )$ if, and only if, the propositional formula ${\left[{\hat{M}}^{\mathrm{tr}(\varphi ),s^{0}}\right]}_{k}\wedge {\left[\mathrm{tr}(\varphi )\right]}_{k}$ is satisfiable.*


The proof of the above theorem is analogous to the proofs presented in [26,53].

## 5. Experimental Results

In this section, we experimentally evaluate the performance of our new translation (We performed our experimental results on a computer equipped with I7-3770 processor, 32 GB of RAM, and the operating system Arch Linux. All the benchmarks together with instructions on how to reproduce our experimental results can be found at the web page https://tinyurl.com/satbmc4dtta-emtl, accessed on 3 November 2022). Our SAT-based BMC algorithm is implemented as a standalone program written in the programming language C++. We compared the new method with the corresponding one from [26]. For both methods, we used the state-of-the-art Kissat SAT solver [54]. We conducted the experiments using the slightly modified TGPP [26], the TTCS [26], and TDPP, and we compared our result with the results generated using the implementation from [26].

### 5.1. Timed Dining Philosophers

As the first benchmark, we used the well-known dining philosophers problem [55], and we extended it using clocks. The system consists of *n* discrete timed automata, each of which models a philosopher, together with *n* automata, each of which models a fork, together with one automaton which models the lackey. The latter automaton is used to coordinate the philosophers’ access to the dining room. In fact, this automaton ensures that no deadlock is possible. The global system is obtained as the parallel composition of the components, which are shown in Figure 4.

We assume that one unit of time represents 30 min. A philosopher has to think at least 30 min (1 time unit, $x_{j}\geq T_{2}$) and at most 2 h and 30 min (5 time units $x_{j}\leq T_{1}$). He also has to eat for, at most, one hour (2 time units, $x_{j}\leq E_{1}$ ), but he also can finish eating earlier ($x_{j}\geq E_{2}$).

Let us consider the following formulae:$\varphi_{1}=F_{\left[0,T_{2}+E_{2}+1\right)}\left({⋁}_{j=1}^{n}Lf\;realeased_{j}\right)$. At least one philosopher will eventually eat and put down both forks.$\varphi_{2}=F_{\left[0,T_{2}+1\right)}\left({⋀}_{j=\{1,3,5,\ldots \}}^{n-1}Eating_{j}\right)$. Eventually, every second philosopher (starting with the first one) eats.$\varphi_{3}=G(F_{\left[0,T_{2}+1\right)}\left({⋁}_{j=1}^{n}Lf\;realeased_{j}\right)$. Every second philosopher (starting with the first one) always eats in the end.

All these formulae are existentially valid in the model of TDPP.

Figure 5 shows experimental results for $\varphi_{1}$ and $\varphi_{2}$. For the simple *eventually* formula $\varphi_{1}$ we can observe that time usage for the method based on the old translation is better than for the method based on the new one. However there is a noticeable difference in memory usage. In this case, the new method is better. For the formulae $\varphi_{2}$ and $\varphi_{3}$, we can see the advantages of the new method.

Figure 6 shows experimental results for $\varphi_{3}$.

Figure 7 shows generated clauses and variables for $\varphi_{1}$ and $\varphi_{3}$.

### 5.2. Timed Generic Pipeline Paradigm

The TGPP (Figure 8) discrete timed automata model [26] consists of a producer producing data within the time interval ([a,b]) or being inactive, a consumer receiving data within the time interval ([c,d]) or being inactive within the time interval ([g,h]), and a chain of *n* intermediate nodes which can be ready for receiving data within the time interval ([c,d]), processing data within the time interval ([e,f]) or sending data. We assume that a=c=e=g=1 and b=d=f=h=2·n+2, where *n* represents number of nodes in the TGPP.

To compare our experimental results with [26], we tested the TGPP discrete timed automata model on the following MTL formulae that existentially hold in the model of TGPP (*n* is the number of nodes). In the below formulae, we use $prod_{0}$, $prod_{1}$, $cons_{0}$, and $cons_{1}$ for ProdReady, ProdSend, ConsReady, and ConsFree respectively. Moreover, we write G for $G_{\left[0,\infty \right)}$. Let us consider the following formulae:$\varphi_{1}=G\left(prod_{0}\vee cons_{0}\right)$. It states that always either the producer has sent the data or the consumer has received the data.$\varphi_{2}=F_{\left[0,2\cdot n+3\right)}\left(G\left(prod_{1}\vee cons_{1}\right)\right)$. It states that eventually in time less then 2·n+3, it is always the case that the producer is ready to send the data or the consumer has received the data.$\varphi_{3}=G\left(F_{\left[0,2\cdot n+3\right)}\left(cons_{1}\right)\right)$. It states that the Consumer infinitely often eventually receives the data in time less than 2·n+3 units.

All these formulae are existentially valid in the model of TGPP.

Charts in Figure 9 show the total time usage and total memory usage for TGPP needed for verification $\varphi_{1}$ and $\varphi_{2}$. In both cases, the new method outperforms the old one. For $\varphi_{2}$, the ${\mathrm{LTL}}_{q}$-based method was able to verify the system with 19 nodes, and the HLTL-based method was able to verify the system only with 9 nodes. For $\varphi_{3}$, the memory usage is similar in both cases. However, the time usage for the old method exponentially grows. The second plot shows the number of generated clauses and variables.

Charts in Figure 10 shows the total time usage and total memory usage for TGPP needed for verification $\varphi_{3}$. The second plot shows the number of generated clauses and variables.

### 5.3. Timed Train Controller System

The TTCS (Figure 11) consists of *n* (for n≥2) trains $T_{1},\ldots ,T_{n}$, each one using its own circular track for traveling in one direction and containing its own clock $x_{i}$, together with controller *C* used to coordinate the access of trains to the tunnel through which all trains have to pass at a certain point. There is only one track in the tunnel, so trains arriving from each direction cannot use it in the same time. There are signals on both sides of the tunnel, which can be either red or green. All trains notify the controller when they request entry to the tunnel or when they leave the tunnel. The controller controls the color of the displayed signal, and the behavior of the scenario depends on the values δ and Δ (Δ>δ+1 makes it incorrect—the mutual exclusion does not hold).

Controller *C* has n+1 locations, with the location 0 being the initial one. The action $Start_{i}$ of train $T_{i}$ denotes the passage from the location away to the location where the train wishes to obtain access to the tunnel. This is allowed only if controller *C* is in location 0. Similarly, train $T_{i}$ synchronizes with controller *C* on action $approach_{i}$, which denotes setting *C* to location i>0, as well as $out_{i}$, which denotes setting *C* to location 0. Finally, action $in_{i}$ denotes the entering of train $T_{i}$ into the tunnel.

Moreover, we assume the following set of propositional variables: $\mathrm{AP}=\{tunnel_{1},\cdots ,$$tunnel_{n}\}$.

Let us consider the following formulae:$\varphi_{1}=F_{\left[0,2\cdot \delta +4\right)}\left({⋁}_{i=1}^{n-1}{⋁}_{j=i+1}^{n}\right)\left(tunnel_{i}\wedge tunnel_{j}\right)$. It expresses that the system violates the mutual exclusion property.$\varphi_{2}=G\left(F_{\left[0,2\cdot \delta +1\right)}tunnel_{1}\right)$. It expresses that the first train can infinitely often and from any state enter the tunnel in time less than 2·δ+1.$\varphi_{3}=G\left(F_{\left[0,2\cdot \delta +7\right)}tunnel_{1}\wedge F_{\left[0,2\cdot \delta +7\right)}\neg tunnel_{1}\right)$. It expresses that the first train is infinitely often in the tunnel and outside the tunnel in time less than 2·δ+7.

All these formulae are existentially valid in the model of TTCS.

As we can see in Figure 12, Figure 13 and Figure 14, the new method surpasses the old one. As we expected, the difference between the two methods is smaller for the simple formula that expresses reachability problem ($\varphi_{1}$). However, a significant difference can be seen for the formulae $\varphi_{2}$ and $\varphi_{3}$. Figure 14 also shows the number of clauses and variables for the new and the old method. As we can see, the numbers of variables and clauses grow exponentially for the old method.

## 6. Statistics

We performed one- and two-sided Wicoxon tests for DPTT (Figure 15). Tests showed that the new method outperforms the old one: the new method used less time (p=0.36), and used less memory (p=0.99).

We performed the two-sided and one-sided Wilcoxon tests for all the experiments. As a dataset, we took the whole set of the experimental results (note that we deleted some results in the figures in Section 5 to make them clear—whole data can be found in the .tar.xz file we delivered).

## 7. Conclusions

In this work, we proposed a new SAT-based BMC for soft real-time systems modeled by discrete time automata with digital clocks and for properties expressible in metric temporal logic with semantics over discrete time automata with digital clocks.

The first step of this method is translating the existential model-checking problem for MTL into the existential model-checking problem for ${\mathrm{LTL}}_{q}$ logic by replacing temporal operators with intervals (MTL) with temporal operators and new propositional variables corresponding to these intervals (${\mathrm{LTL}}_{q}$). The second step is translating the existential model-checking problem for ${\mathrm{LTL}}_{q}$ into the satisfiability problem for the propositional formulae. The efficiency of the new method is due to the fact that only one additional clock for measuring global time is needed, unlike the earlier method [26], which translates the existential model-checking problem for MTL into the existential model-checking problem translation to HLTL.

The earlier method [26] needs to add to a timed automaton one extra clock, one extra path, and an extra transition for each occurrence of the temporal operator in the formula.

We implemented our method as a standalone program written in the programming language C++. This implementation allowed us to experimentally evaluate and compare the new approach with the old one.

The experimental results show that our approach is significantly better than the approach based on translation to HLTL. The new method substantially reduces the conjunction normal form (CNF) formula’s size, an input formula for the SAT solver. The reduced size of the CNF formula causes the SAT solver to use much less time and memory to determine the satisfiability of the input formula.

In future work, we plan to extend our method by adding discrete data [56]. We also would like to improve and develop and prove the method presented in [49].

## Figures and Tables

**Figure 1 sensors-22-09552-f001:**
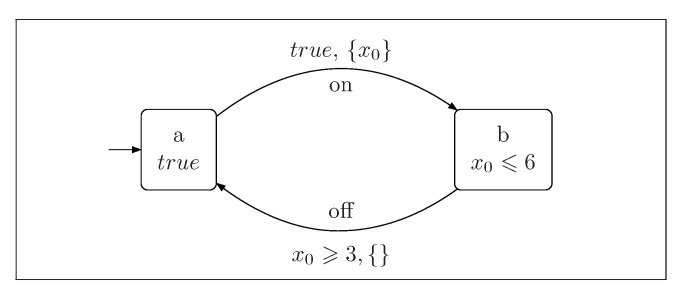
The simple light switch.

**Figure 2 sensors-22-09552-f002:**
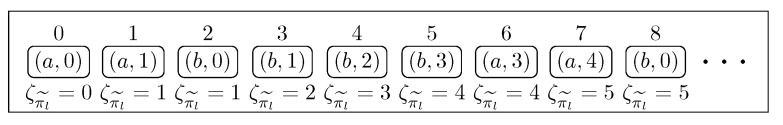
An example of the path.

**Figure 3 sensors-22-09552-f003:**
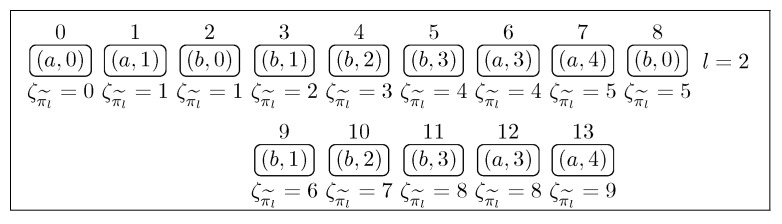
An example of the *k*-path, which is a loop.

**Figure 4 sensors-22-09552-f004:**
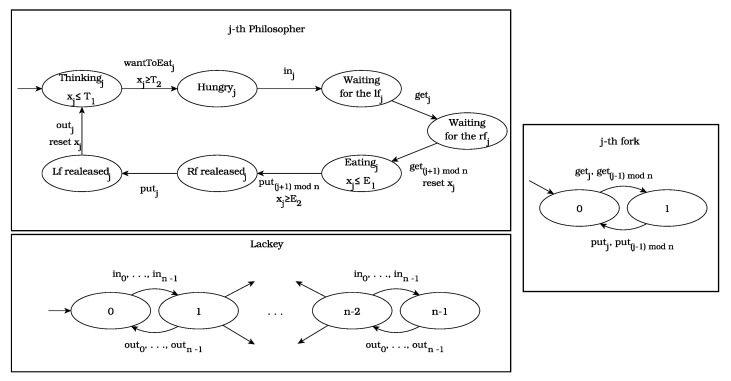
The TDPP system.

**Figure 5 sensors-22-09552-f005:**
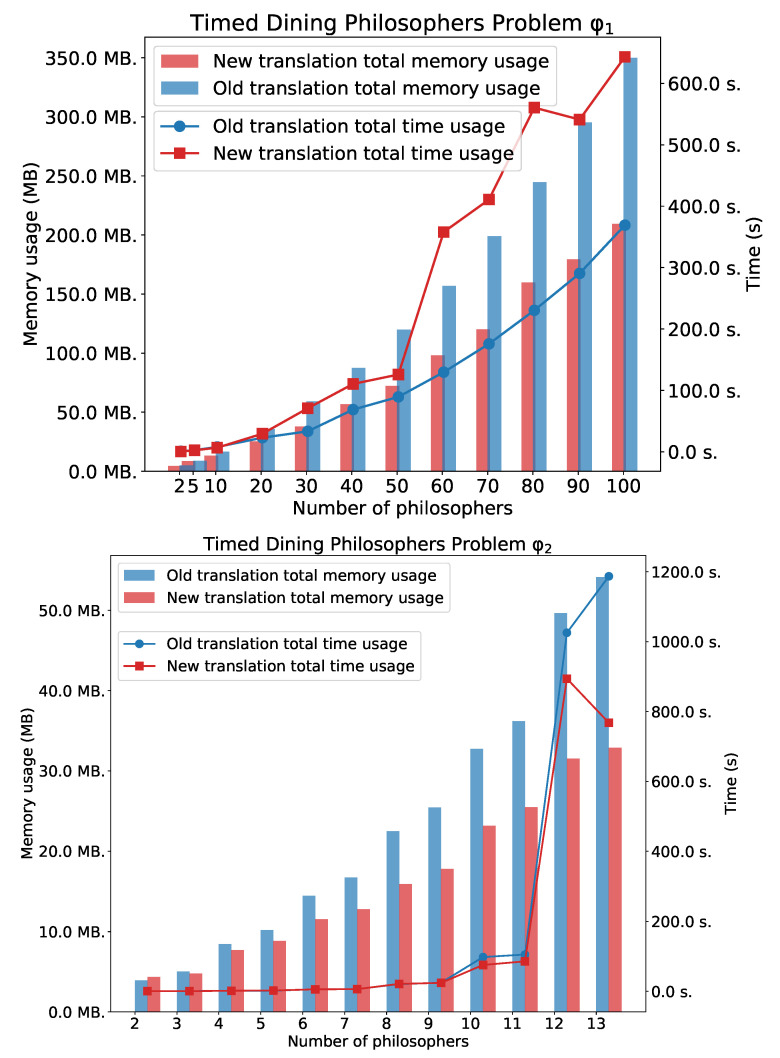
TDPP with *n* philosophers: $\varphi_{1}$ and $\varphi_{2}$.

**Figure 6 sensors-22-09552-f006:**
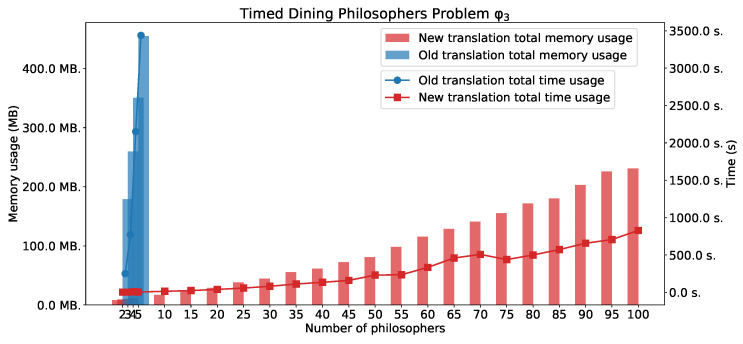
The TDPP with *n* philosophers: $\varphi_{3}$.

**Figure 7 sensors-22-09552-f007:**
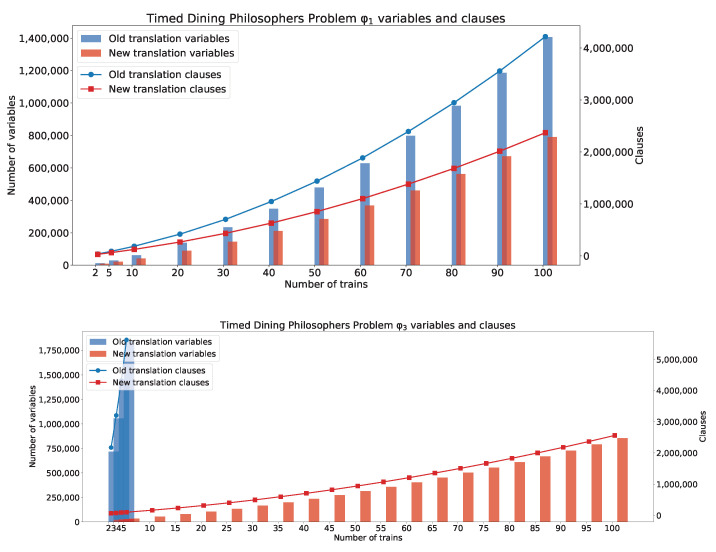
The TDPP with *n* philosophers clauses and variables: $\varphi_{1}$ and $\varphi_{3}$.

**Figure 8 sensors-22-09552-f008:**
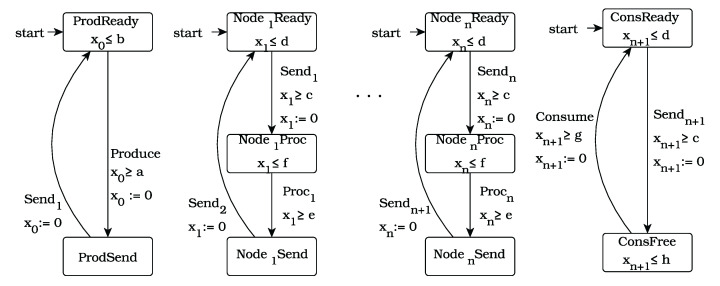
The TGPP system.

**Figure 9 sensors-22-09552-f009:**
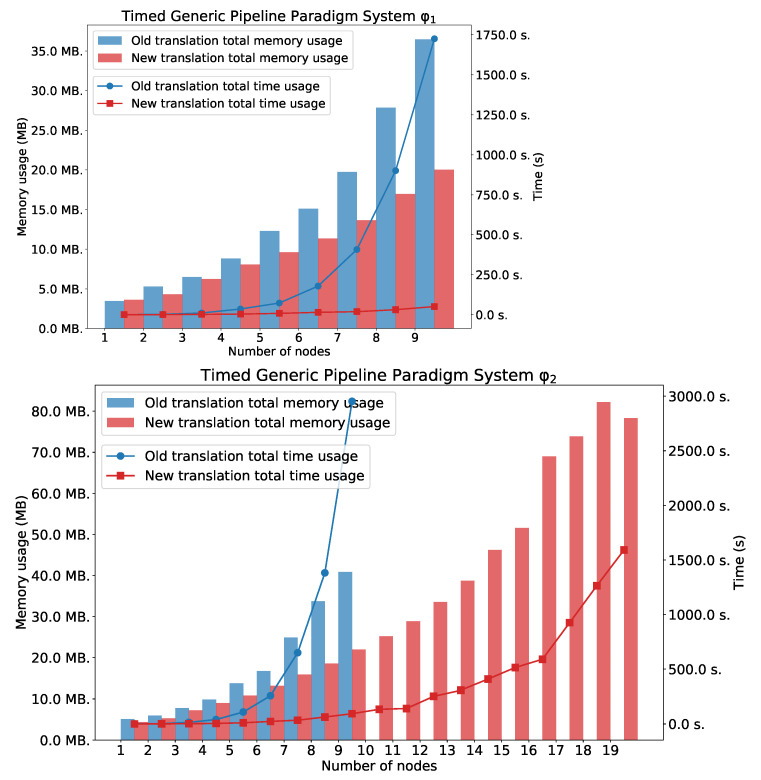
$\varphi_{1}$ and $\varphi_{2}$: TGPP with *n* nodes.

**Figure 10 sensors-22-09552-f010:**
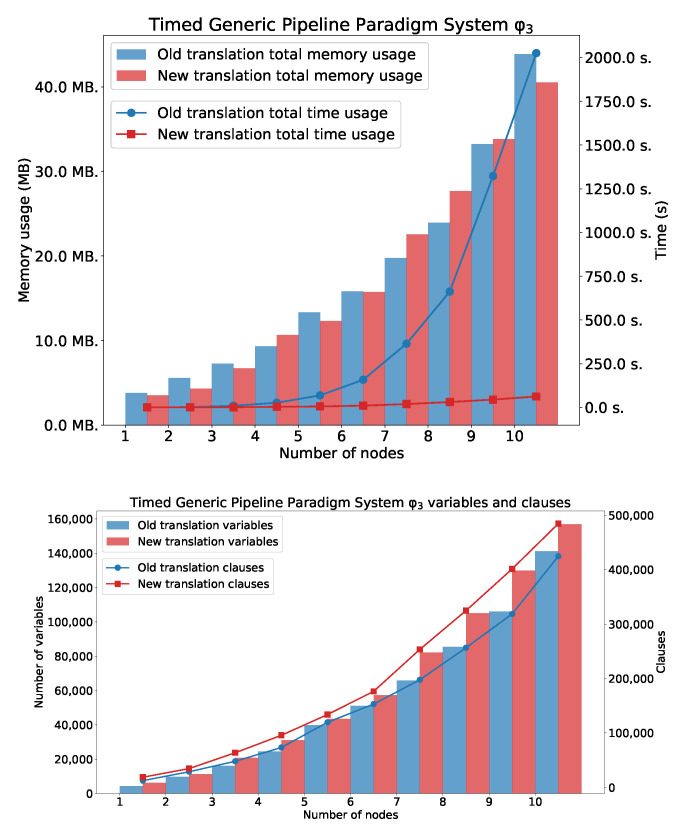
Results for $\varphi_{3}$: TGPP with *n* nodes. Number of variables and clauses for $\varphi_{3}$.

**Figure 11 sensors-22-09552-f011:**
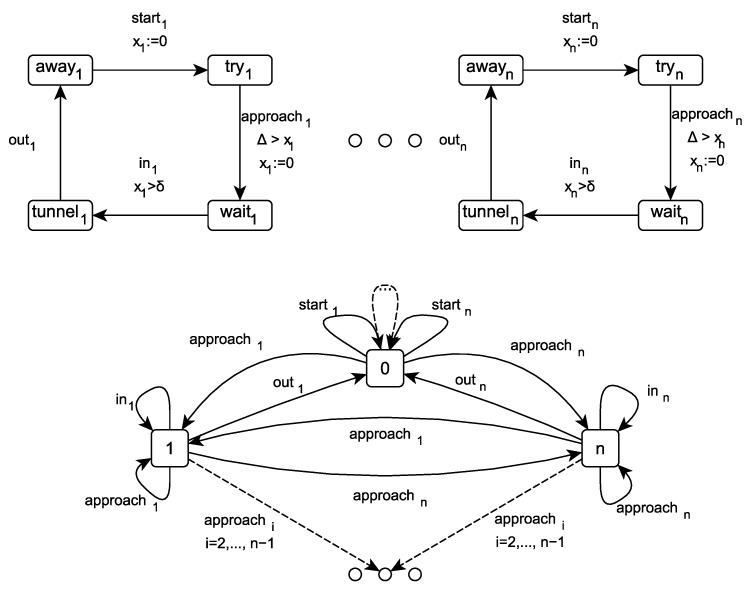
The TTCS system.

**Figure 12 sensors-22-09552-f012:**
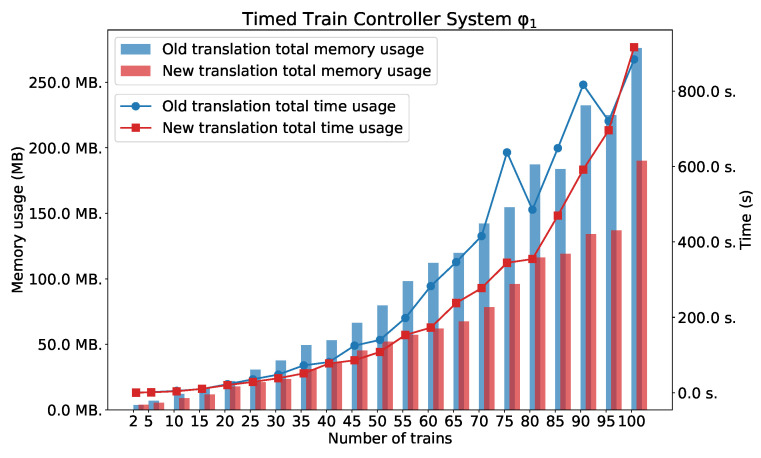
$\varphi_{1}$: TTCS with *n* trains.

**Figure 13 sensors-22-09552-f013:**
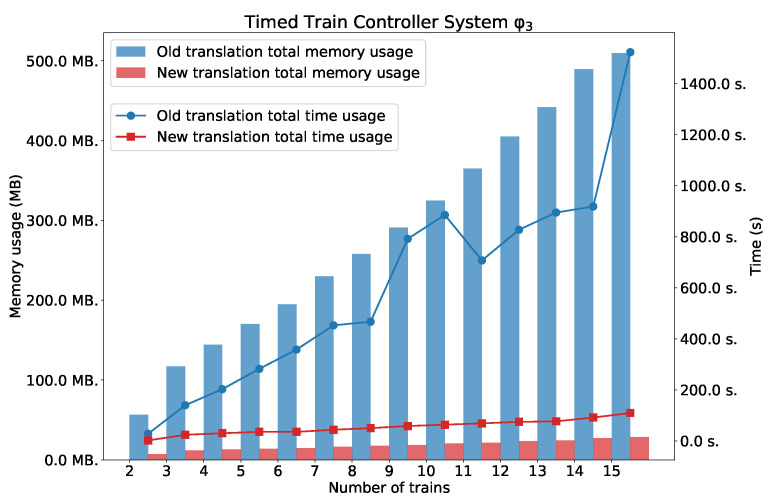
$\varphi_{3}$: TTCS with *n* trains.

**Figure 14 sensors-22-09552-f014:**
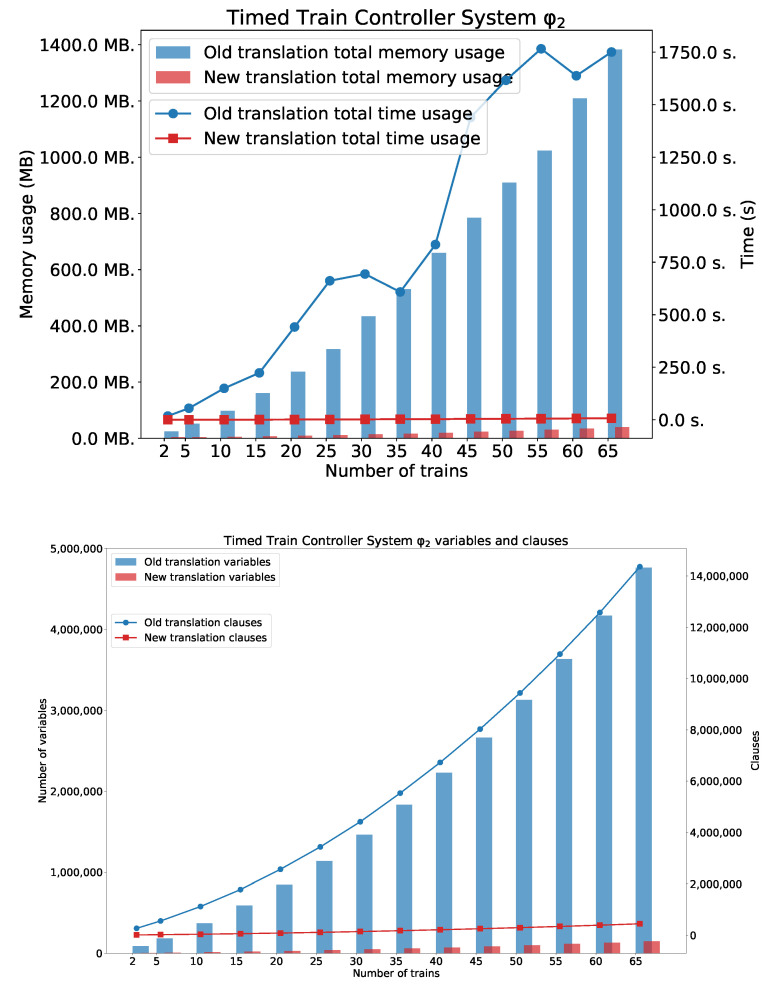
Results for $\varphi_{2}$. Number of variables and clauses for $\varphi_{2}$ and TTCS with *n* trains.

**Figure 15 sensors-22-09552-f015:**
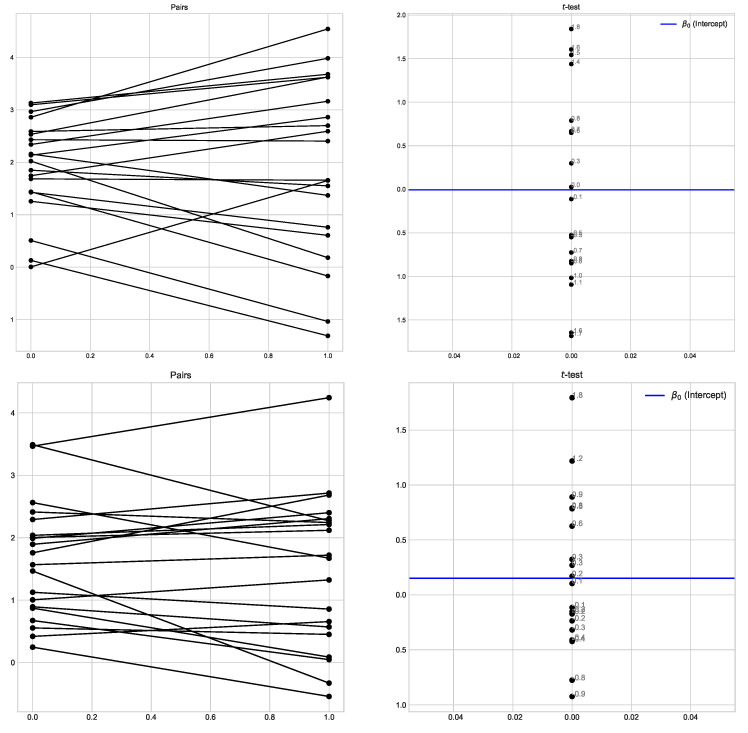
TDPP: Pairs Wilcoxon plots for total time usage and total memory usage for $\varphi_{1}$, $\varphi_{2}$, and $\varphi_{3}$.

## Data Availability

Not applicable.

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
