# Peer review of "Bounded Model Checking for Metric Temporal Logic Properties of Timed Automata with Digital Clocksâ€"

_sensors, 2022, doi:10.3390/s22239552_

Round 1
Reviewer 1 Report
The authors of the paper describe their proposed approach for Bounded Model Checking for Metric Temporal Logic properties of Timed Automata with Digital Clocks. The topic is interesting and with possible future applicability. However, the paper needs several improvements:
1) the main contribution and originality should be explained in more detail, which part of the proposal is new?
2) the motivation of the approach needs further clarification, why this research was undertaken?
3) discussion of related work in type 2 should be expanded with more recent work
4) Minor grammar and syntax issues need correction
5) more examples to illustrate the proposed ideas are needed
6) the conclusions should be extended with more discussion of future works
7) More references to recent related papers could be included (only 37 references now in the paper)
8) The authors contributions were not disclosed
9) Please define all variables and parameters in the equations
Author Response
Dear Sir or Madame,
We want to thank you for your valuable comments.
We have attached our answers and the new version of the manuscript. In the manuscript file, we sign the changes we have made in red.
1) the main contribution and originality should be explained in more detail, which part of the proposal is new?
-- We added the list with the paper's main contributions and pointed out the originality of the article (lines 15-22 and 70-84).
2) the motivation of the approach needs further clarification, why this research was undertaken?
-- We added a short description of why this research was undertaken by us (lines 61-69).
3) discussion of related work in type 2 should be expanded with more recent work
-- We extended the Related work section by adding new articles. We also expanded the description of the gap translation (lines 126-130).
4) Minor grammar and syntax issues need correction
We checked the paper and corrected the grammar and syntax issues.
5) more examples to illustrate the proposed ideas are needed
-- We reorganise the examples section.
6) the conclusions should be extended with more discussion of future works
-- We extended the Conclusions section by adding information about future work. We plan to expand our abstract from the CS&P conference by adding each translation step, proof, and more experiments. We would also like to enrich the discrete timed model using discrete variables.
7) More references to recent related papers could be included (only 37 references now in the paper)
-- We extended the Related work section by adding new articles.
8) The authors contributions were not disclosed
-- We fixed it.
9) Please define all variables and parameters in the equations
-- We fixed it.

Reviewer 2 Report
The article is excellent in every respect. Only the literature review section is deficient. In the literature review section, authors should extend and introduce a detailed comparative study so that the reader find the originality of the proposed work compared with other relevant studies.
Author Response
Dear Sir or Madame,
We want to thank you for your valuable comments.
In the literature review section, authors should extend and introduce a detailed comparative study so that the reader find the originality of the proposed work compared with other relevant studies.
-- We extended the Related work section by adding new articles. We also pointed out the originality of the paper in the Introduction section.

Reviewer 3 Report
The paper is interesting and well-written.
Major comments
1. Other discrete time model checking methods should be referred to, for example:
CCTL, based on time intervals in discrete-time systems: https://doi.org/10.1023/A:1024437214071
discrete-time CTL: (DTCTL for embedded systems) https://doi.org/10.1016/S1474-6670(17)30171-4,
durational transition graphs (DTGs) doi:10.1016/j.tcs.2005.11.020
HyperMTL: https://doi.org/10.1007/978-3-030-55754-6_18
2. In my opinion, not so much space should be devoted to the comparison with another implementation, if the previous one [28] is from the same authors.
3. TTCS is not presented in [28]. A physical figure of the trains' interaction is needed.
4. The lack of communication variables in the model (e.g. introduced in Uppaal) is a significant limitation in the application of the method. Rather, only simple systems can be modeled without variables. This serious limitation should be stated in the conclusions.
5. Parallel composition should be defined.
Minor remarks
1. Page 3:\delta should have a name for references to it
2. Page 4: Time transition: shouldn't it be 0 \lt δ' \le δ ?
3. Page 4: p \in AP, p is a variable, AP is a set of propositions. Is it ok?
4. Page 4: I should be defined "let..." not referred to "where..."
5. Page 4: Please comment: is \Gamma a set of indices of time transitions?
6. Page 4: Please comment: is \zeta a sum of time intervals in \rho?
7. Page 5: Why is a model called abstract?
8. Page 5: The abstract state is not defined. Whi is it called abstract?
9. Page 8: I think that a space is needed between E and tr (here and below).
10. Page 12: a line break required between [\alpha \wedge \beta]... and [\alpha \vee \beta]

Author Response
Dear Sir or Madame,
We want to thank you for your valuable comments.
1. Other discrete time model checking methods should be referred to, for example:
CCTL, based on time intervals in discrete-time systems: https://doi.org/10.1023/A:1024437214071
discrete-time CTL: (DTCTL for embedded systems) https://doi.org/10.1016/S1474-6670(17)30171-4,
durational transition graphs (DTGs) doi:10.1016/j.tcs.2005.11.020
HyperMTL: https://doi.org/10.1007/978-3-030-55754-6_18
--- We added those articles and others in the related work section.
2. In my opinion, not so much space should be devoted to the comparison with another implementation, if the previous one [28] is from the same authors.
--- We left the comparison because other reviewers did not ask to delete it. Our research is based on a completely different method, which is less effective, and thus we wanted to show the validity and efficiency of our new approach.
3. TTCS is not presented in [28]. A physical figure of the trains' interaction is needed.
-- The figure is added in the first version of the paper (Fig 11). We deleted this citation.
4. The lack of communication variables in the model (e.g. introduced in Uppaal) is a significant limitation in the application of the method. Rather, only simple systems can be modeled without variables. This serious limitation should be stated in the conclusions.
-- The main point of our research was to develop the method corresponding to the one presented in [26]. However, in future work, we plan to expand the technique by adding discrete variables like in the paper [59].
5. Parallel composition should be defined.
-- We added the definition of the parallel composition.
We also applied all the suggestions from the "Minor remarks" section:
Minor remarks
1. Page 3:\delta should have a name for references to it
-- We added this.
2. Page 4: Time transition: shouldn't it be 0 \lt δ' \le δ ?
-- We fixed this.
3. Page 4: p \in AP, p is a variable, AP is a set of propositions. Is it ok?
-- We fixed this.
4. Page 4: I should be defined "let..." not referred to "where..."
-- We fixed this.
5. Page 4: Please comment: is \Gamma a set of indices of time transitions?
-- Yes, it is.
6. Page 4: Please comment: is \zeta a sum of time intervals in \rho?
-- It is the sum of time passage of time transitions.
7. Page 5: Why is a model called abstract?
-- It is due to the fact that the $\simeq$ is an equivalence relation. Such a relation is also called time abstract.
8. Page 5: The abstract state is not defined. Whi is it called abstract?
-- The states of the abstract model are called abstract states. The definition can be found in line 298.
9. Page 8: I think that a space is needed between E and tr (here and below).
-- We checked it; however, it looked bad.
10. Page 12: a line break required between [\alpha \wedge \beta]... and [\alpha \vee \beta]
-- We fixed this.

Round 2
Reviewer 1 Report
The authors have addressed all my concerns and the paper can be accepted.